# BEYOND QUANTIZATION: POWER AWARE NEURAL NETWORKS

## ABSTRACT

Power consumption is a major obstacle in the deployment of deep neural networks (DNNs) on end devices. Existing approaches for reducing power consumption rely on quite general principles, including avoidance of multiplication operations and aggressive quantization of weights and activations. However, these methods do not take into account the precise power consumed by each module in the network, and are therefore far from optimal. In this paper we develop accurate power consumption models for all arithmetic operations in the DNN, under various working conditions. Surprisingly, we reveal several important factors that have been overlooked to date. Based on our analysis, we present PANN (power-aware neural network), a simple approach for approximating any full-precision network by a low-power fixed-precision variant. Our method can be applied to a pre-trained network, and can also be used during training to achieve improved performance. In contrast to previous methods, PANN incurs only a minor degradation in accuracy w.r.t. the full-precision version of the network, even when working at the power-budget of a 2-bit quantized variant. In addition, our scheme enables to seamlessly traverse the power-accuracy trade-off at deployment time, which is a major advantage over existing quantization methods that are constrained to specific bit widths.

## 1 INTRODUCTION

With the ever increasing popularity of deep neural networks (DNNs) for tasks like face detection, voice recognition, and image enhancement, power consumption has become one of the major considerations in the design of DNNs for resource-limited end-devices. Over the last several years, a plethora of approaches have been introduced for achieving power efficiency in DNNs. These range from specialized architectures (Sandler et al., 2018; Huang et al., 2019; Tan et al., 2019; Radosavovic et al., 2020), to hardware oriented methods like multiplier-free designs and low-precision arithmetic.

Multiplier aware methods attempt to reduce power consumption by avoiding the costly multiplication operations, which dominate the computations in a DNN. Several works replaced multiplications by additions (Courbariaux et al., 2015; Li et al., 2016; Chen et al., 2020) or by bit shift operations (Elhoushi et al., 2019) or both (You et al., 2020). Others employed efficient matrix multiplication operators (Tschannen et al., 2018; Lavin & Gray, 2016). However, most methods in this category introduce dedicated architectures, which require training the network from scratch. This poses a severe limitation, as different variants of the network need to be trained for different power constraints.

Low-precision DNNs reduce power consumption by using low-precision arithmetic. This is done either via quantization-aware training (QAT) or with post-training quantization techniques. The latter avoid the need for retraining the network but often still require access to a small number of calibration samples in order to adapt the network's weights. Such techniques include approaches like re-training, fine-tuning, calibration and optimization (Banner et al., 2019; Jacob et al., 2018; Nahshan et al., 2019; Li et al., 2021). All existing methods in this category suffer from a large drop in accuracy with respect to the full-precision version of the network, especially when working at very low bit widths. Moreover, similarly to the multiplier-free approaches, they do not provide a mechanism for traversing the power-accuracy trade-off without actually changing the hardware (*e.g.*, replacing an 8-bit multiplier by a 4-bit one). In this work, we introduce a *power-aware neural network* (PANN) approach that allows to dramatically cut down the power consumption of any model. Our method can be applied at post-training to improve the power efficiency of a pre-trained model, or in a QAT setting

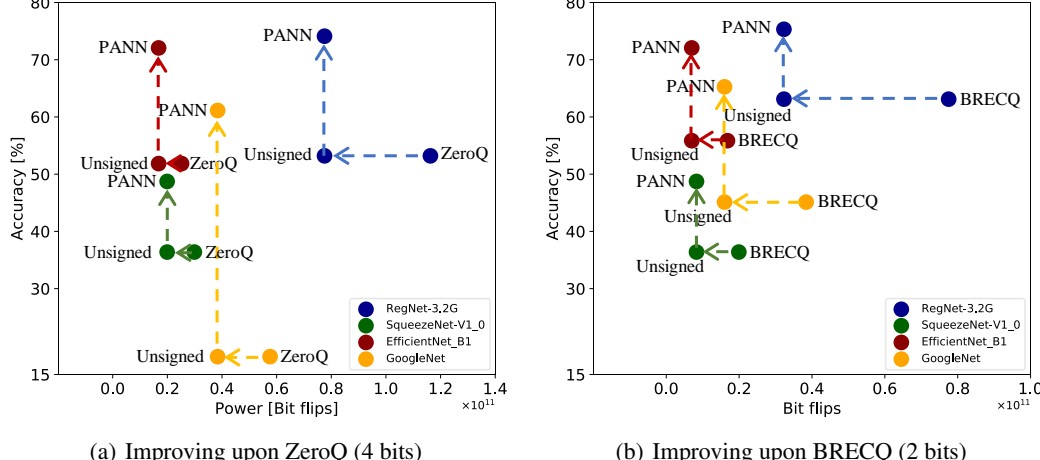

Figure 1: **Power-accuracy trade-off at post training**. For each pre-trained full-precision model, we used (a) ZeroQ (Cai et al., 2020) and (b) BRECQ (Li et al., 2021) to quantize the weights and activations at post-training. In (a) we quantize to 4 bits and in (b) to 2 bits. Then, we convert the quantized models to work with unsigned arithmetic ($\leftarrow$), which already cuts down 33% of the power consumption in (a) and 58% in (b) (assuming a 32 bit accumulator). Using our PANN approach to quantize the weights (at post-training) and remove the multiplier, further decreases power consumption and allows achieving higher accuracy for the same power level ($\uparrow$). See more examples in Appendix A.5.1

to obtain even improved results. Our approach is based on careful analysis of the power consumed by additions and multiplications, as functions of several factors. We rely on bit toggling activity, which is known to be the main factor affecting dynamic power consumption, and support our theoretical analysis with accurate gate-level simulations on a 5nm process.

Our first important observation is that a major portion of the power consumed by a DNN is due to the use of signed integers. We therefore present a simple method for converting any pre-trained model to use unsigned arithmetic. This conversion does not change the functionality of the model and, as can be seen in Fig. 1, dramatically reduces power consumption on common hardware configurations.

Our second observation is that the multiplier's power consumption is dominated by the larger bit width among its two inputs. Therefore, although high accuracy can often be achieved with quite drastic quantization of only the weights, this common practice turns out to be ineffective in terms of power consumption. To be able to take advantage of drastic weight quantization, here we introduce a method that allows removing the multiplier altogether. Our approach can work in combination with any activation quantization method. We show theoretically and experimentally that this method is far advantageous over existing quantization methods at low power budgets, both at post-training and in QAT settings (see Fig. 1 and Sec. 6).

Our method allows working under any power constraint by tuning the number of additions used to approximate each mutltiply-accumulate (MAC) operation. This is in contrast to regular quantization methods, which are limited to particular values. We can thus traverse the power-accuracy trade-off without changing the architecture (*e.g.*, bit width of the multiplier), as required by existing methods.

## 2 RELATED WORK

**Avoiding multiplications** In fixed point (integer) representation, additions are typically much more power-efficient than multiplications (Horowitz, 2014b;a). Some works suggested to binarize or ternarize the weights to enable working with additions only (Courbariaux et al., 2015; Lin et al., 2015; Li et al., 2016). However, this often severely impairs the network's accuracy. Recent works suggested to replace multiplications by bit shifts (Elhoushi et al., 2019) or additions (Chen et al., 2020) or both (You et al., 2020). Other methods reduce the number of multiplications by inducing

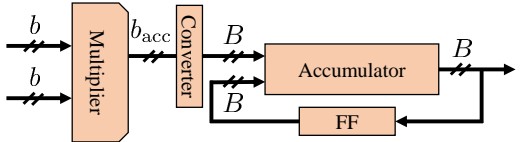

Figure 2: **Multiply-accumulate.** The multiplier accepts two $b$-bit inputs. The product is then summed with the previous $B$ bit sum, which awaits in the flip-flop (FF) register.

| Element | toggles |
|---|---|
| Multiplier inputs | $0.5b+0.5b$ |
| Multiplier's internal units | $0.5b^2$ |
| Accumulator input | $0.5B$ |
| Accumulator sum & FF | $0.5b_{\text{acc}}+0.5b_{\text{acc}}$ |

Table 1: **Average number of bit flips per signed MAC.** The $b$-bit multiplier inputs are drawn uniformly from $[-2^{b-1}, 2^{b-1})$ and its $b_{\text{acc}} = 2b$ bit output is summed with the $B$-bit number in the FF.

sparsity (Venkatesh et al., 2016; Mahmoud et al., 2020), decomposition into smaller intermediate products (Kim et al., 2016), Winograd based convolutions (Lavin & Gray, 2016), or Strassen's matrix multiplication algorithm (Tschannen et al., 2018). Some of these methods require internal changes in the model, a dedicated backpropagation scheme, or other modifications to the training process.

**Quantization**  DNN quantization approaches include post-training quantization (PTQ), which is applied to a pre-trained model, and quantization-aware training (QAT), where the network's weights are adapted to the quantization during training (Gupta et al., 2015; Louizos et al., 2018; Achterhold et al., 2018; Esser et al., 2019). PTQ methods are more flexible in that they do not require access to the training set. These methods show optimal results for 8-bit quantization, but tend to incur a large drop in accuracy at low bit widths. To battle this effect, some PTQ methods minimize the quantization errors of each layer individually by optimizing the parameters over a calibration set (Nahshan et al., 2019; Nagel et al., 2020; Hubara et al., 2020). Others use nonuniform quantization (Liu et al., 2021; Fang et al., 2020). Effort is also invested in avoiding the need of any data sample for calibration (Cai et al., 2020; Shoukai et al., 2020; Nagel et al., 2019; Haroush et al., 2020). These methods, however, still show a significant drop in accuracy at the lower bit widths, while frequently requiring additional computational resources. Common to all quantization works is that they lack analysis of the power consumed by each arithmetic operation as a function of bit-width, and thus cannot strive for optimal power-accuracy trade-offs.

## 3 POWER CONSUMPTION OF A CONVENTIONAL DNN

The total amount of power consumed by a logic circuit can be attributed to two main sources: a static power component and a dynamic one. The static power is due to a constant leakage current. It does not depend on the circuit's activity and is typically the smaller component among the two. The dynamic power consumed by each node in the circuit is given by $P = CV^2 f\alpha$, where $C$ is the node capacitance, $V$ is the supply voltage, $f$ is the operating frequency, and $\alpha$ is the switching activity factor (the average number of bit flips per clock) (Nasser et al., 2017). Here we focus on dynamic power, which is a major contributor to the overall power consumption (see Appendix A.1 and (Karimi et al., 2019; Kim et al., 2020)). Also, this is the only factor affected by the DNN architecture.

Most of the computation in a forward pass of a DNN can be attributed to MAC operations. As shown in Fig. 2, MACs involve a multiplier that accepts two $b$-bit numbers and outputs a $b_{\text{acc}}$-bit result ($b_{\text{acc}} = 2b$ to account for the largest possible product), and an accumulator with a large bit width $B$ to which the multiplier's output is added repeatedly.

To understand how much power each of these components consumes, we simulated them in Python. For the multiplier, we used the Booth-encoding architecture, which is considered efficient in terms of bit toggling (Asif & Kong, 2015). For the accumulator, we simulated a serial adder. Our Python simulation allows measuring the total number of bit flips in each MAC operation, including at the inputs, at the outputs, in the flip-flop (FF) register holding the previous sum, and within each of the internal components (*e.g.*, the full-adders) of the multiplier. We also verified our analysis with an accurate physical gate-level simulation on a 5nm process and found good agreement in terms of the dependence of power consumption on the bit widths (see details in Appendix A.1).

Table 1 shows the average number of bit flips per MAC when both inputs to the multiplier are drawn uniformly at random from $[-2^{b-1}, 2^{b-1})$ (Gaussian inputs lead to similar results; please see

Appendix Figs 7-8). As can be seen, the power consumed by the multiplier is given by[1]

$$P_{\text{mult}} = 0.5b^2 + b, \tag{1}$$

where $0.5b^2$ is due to the bit toggling in the internal units, and $0.5b$ is contributed by the bit flips in each input. The power consumed by the accumulator is given by

$$P_{\text{acc}} = 0.5B + 2b, \tag{2}$$

where $0.5B$ is due to the bit toggling in its input coming from the multiplier, $0.5b_{\text{acc}} = b$ (recall $b_{\text{acc}} = 2b$) to the bit flips at the output, and an additional $0.5b_{\text{acc}} = b$ to the bit flips in the FF. These results lead us to our first important observation.

**Observation 1.** *A dominant source of power consumption is the bit toggling at the input of the accumulator ($0.5B$).*

Suppose, for example, we use $b = 4$ bits for representing the weights and activations and employ a $B = 32$ bit accumulator, as common in modern architectures (Kalamkar et al., 2019; Rodriguez et al., 2018). Then the toggling at the input of the accumulator ($0.5B = 16$) is responsible for $44.4\%$ of the total power consumption ($P_{\text{mult}} + P_{\text{acc}} = 36$). At lower bit widths, this percentage is even larger.

Unfortunately, existing quantization methods and multiplier-free designs do not battle this source of power consumption. Ni et al. (2021) have recently shown that the bit-width $B$ of the accumulator can be somewhat reduced by explicitly accounting for overflows. However, this approach requires dedicated training, and degrades the network's classification accuracy at low values of $B$. As we now show, it is possible to drastically reduce the bit toggles at the input of the accumulator at post-training *without changing the model's functionality* (thus retaining the same classification accuracy).

## 4    SWITCHING TO UNSIGNED ARITHMETIC

Since the output of the multiplier has only $2b$ bits, one could expect to experience no more than $b$ bit flips on average at the accumulator's input. Why do we have $0.5B$ bit flips instead? The reason is rooted in the use of signed arithmetic. Specifically, negative numbers are represented using two's complement, and thus switching between positive and negative numbers results in flipping of many of the higher bits. For example, when using a 32 bit accumulator, if the output of the multiplier switches from $+2$ to $-2$, then the bits at the input of the accumulator switch from 00000000000000000000000000000010 to 11111111111111111111111111111110. Note that this effect is dominant only at the accumulator's input simply because sign changes at the output are rare.

If we could work with unsigned integers, then the higher bits at the accumulator's input would always remain zero, which would lead to a substantial reduction in power consumption without any performance degradation. To quantify this, we repeated the experiment of Sec. 3, but with the $b$-bit inputs to the multiplier now drawn uniformly from $[0, 2^{b-1})$ (see Appendix A.2 for details). In this case, the average number of bit flips at the input of the accumulator reduced from $0.5B$ to $0.5b_{\text{acc}} = b$. Specifically, the average power consumption of an unsigned MAC operation was measured to be

$$P_{\text{mult}}^{\text{u}} = 0.5b^2 + b \tag{3}$$

due to the multiplier and

$$P_{\text{acc}}^{\text{u}} = 3b \tag{4}$$

due to the accumulator. In (4), $2b$ bit flips occur at the accumulator's output and the FF, and $b$ bit flips occur at the accumulator's input coming from the multiplier. Thus, although the mutliplier's power (3) turns out to be the same as in the signed setting (1), the accumulator's power (4) is substantially reduced w.r.t. the signed case (2).

Converting a pre-trained network with ReLU activation functions to work with unsigned integers is simple. Specifically, consider a layer performing $y = Wx + b$. The elements of $x$ are non-negative because of the preceding ReLU[2]. Therefore, we can split any such layer into two parallel layers as

$$y^+ = W^+x + b^+, \qquad y^- = W^-x + b^-, \tag{5}$$

---

[1]The amount of power consumed by a single bit flip may vary across platforms (*e.g.*, between a 5nm and a 45nm fabrication), but the number of bit flips per MAC does not change. We therefore report power in units of bit-flips, which allows comparing between implementations while disregarding the platform.

[2]Batch-norm layers should first be absorbed into the weights and biases.

where $W^+ = \mathrm{ReLU}(W)$, $b^+ = \mathrm{ReLU}(b)$, $W^- = \mathrm{ReLU}(-W)$, $b^- = \mathrm{ReLU}(-b)$, and compute

$$y = y^+ - y^-. \tag{6}$$

This way, all MACs are converted to unsigned ones in (5), and only a single subtraction per output element is needed in (6). This one subtraction is negligible w.r.t. the MACs, whose number is usually in the thousands. Please see Fig. 11(b) in the Appendix for a schematic illustration.

Figure 1 shows the effect that this approach has on the power consumption of several pretrained networks for ImageNet classification. Figures 1(a) and 1(b) show 4 bit and 2 bit quantized networks, respectively. With a 32 bit accumulator, merely switching to unsigned arithmetic cuts $33\%$ and $58\%$ of the power consumption of these networks (see Appendix A.3.1 for other accumulator bit widths).

## 5    REMOVING THE MULTIPLIER

Having reduced the power consumed by the accumulator, we now turn to treat the multiplier. Quantization methods often use different bit widths for the weights and activations. This flexibility allows achieving good classification accuracy with quite aggressive quantization of one of them (typically the weights), but a finer quantization of the other. An interesting question is whether this approach is beneficial in terms of power consumption.

We repeated the experiment of Sec. 3, this time with the multiplier inputs having different bit widths, $b_w$ and $b_x$. We focused on the standard setting of signed numbers, which we drew uniformly from $[-2^{b_w}, 2^{b_w-1})$ and $[-2^{b_x}, 2^{b_x-1})$. Interestingly, we found that the average number of bit flips in the multiplier's internal units is affected only by the larger among $b_w$ and $b_x$. Accounting also for the bit flips at the inputs, we obtained that the multiplier's total average power is

$$P_{\mathrm{mult}} = 0.5 \left(\max\{b_w, b_x\}\right)^2 + 0.5(b_w + b_x). \tag{7}$$

We found this surprising behavior to be characteristic of both the Booth multiplier and the simple serial multiplier, and verified it also with accurate simulations on a 5nm silicon process gate level synthesis (see Appendix Figs. 9,10). This leads us to our second important observation.

**Observation 2.** *There is marginal benefit in the common practice of decreasing the bit width of only the weights or only the activations, at least in terms of the power consumed by the multiplier.*

It should be noted that in the case of unsigned numbers, where inputs are drawn uniformly from $[0, 2^{b_w-1})$ and $[0, 2^{b_x-1})$, there exists some power save when reducing one of the bit widths, especially for the serial multiplier (see Appendix Fig. 10). This highlights again the importance of unsigned arithmetic. However, in all our experiments we do not take this extra benefit of our approach into account when computing power consumption, so that our reports are conservative.

To benefit from the ability to achieve high precision with drastic quantization of only the weights, we now explore a solution that allows removing the multiplier altogether. Unlike other multiplier-free designs, our method allows converting any full-precision pre-trained model into a low-precision power-efficient one without any changes to the architecture.

### 5.1    POWER AWARE WEIGHT QUANTIZATION

Consider the computation

$$y = \sum_{i=1}^{d} w_i \cdot x_i, \tag{8}$$

which involves $d$ MACs. Here, $\{w_i, x_i\}$ are the weights and activations of a convolution or a fully-connected layer. Given $\{w_i, x_i\}$ in full precision, our goal is to accurately approximate (8) in a power-efficient manner. When quantizing the weights and activations we obtain the approximation

$$y \approx \sum_{i=1}^{d} \gamma_w \mathcal{Q}_w(w_i) \cdot \gamma_x \mathcal{Q}_x(x_i), \tag{9}$$

where the quantizers $\mathcal{Q}_w(\cdot)$ and $\mathcal{Q}_x(\cdot)$ map $\mathbb{R}$ to $\mathbb{Z}$, and $\gamma_w$ and $\gamma_x$ are their quantization steps[3]. To make the computation (9) power efficient, we propose to *implement multiplications via additions*.

---

[3] In quantized models MAC operations are always performed on integers and rescaling is applied at the end.

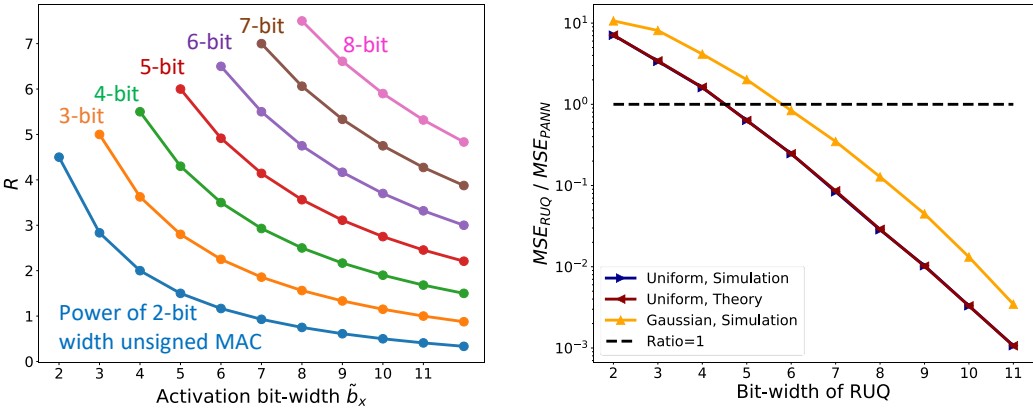

(a) Number of additions vs. bit width in PANN

(b) The ratio between $\text{MSE}_{\text{RUQ}}$ and $\text{MSE}_{\text{PANN}}$

Figure 3: (a) Each color represents the power of an unsigned $b_x$-bit MAC for some value of $b_x$. In PANN, we can move on a constant power curve by modifying the number of additions per element $R$ (vertical axis) on the expense of the activation bit width $\tilde{b}_x$ (horizontal axis). (b) We plot the ratio between the quantization errors of a regular quantizer (RUQ) and a PANN tuned to work at the same power budget. As can be seen, PANN outperforms RUQ at the low bit widths (where the MSE ratio is above 1). It should be noted that at the high bit widths, both approaches achieve low errors in absolute terms, but RUQ is relatively better. In the Gaussian setting, which is closer to the distribution of DNN weights and activations, the range over which PANN outperforms RUQ is a bit larger.

Specifically, assume $\mathcal{Q}_w(w_i)$ is a non-negative integer (as in Sec. 4). Then we can implement the term $\mathcal{Q}_w(w_i) \cdot \mathcal{Q}_x(x_i)$ as

$$\mathcal{Q}_w(w_i) \cdot \mathcal{Q}_x(x_i) = \underbrace{\mathcal{Q}_x(x_i) + \cdots + \mathcal{Q}_x(x_i)}_{\mathcal{Q}_w(w_i) \text{ times}}, \tag{10}$$

so that (9) is computed as

$$y \approx \gamma_w \gamma_x \sum_{i=1}^{d} \sum_{j=1}^{\mathcal{Q}_w(w_i)} \mathcal{Q}_x(x_i). \tag{11}$$

This is the basis for our power-aware neural network (PANN) design.

Let $\boldsymbol{w} = (w_1, \ldots, w_d)^T$ and $\boldsymbol{x} = (x_1, \ldots, x_d)^T$ denote the full precision weights and activations, and denote their quantized versions by $\boldsymbol{w}_q = (\mathcal{Q}_w(w_1), \ldots, \mathcal{Q}_w(w_d))^T$ and $\boldsymbol{x}_q = (\mathcal{Q}_x(x_1), \ldots, \mathcal{Q}_x(x_d))^T$, respectively. Note that as opposed to conventional quantization methods, our approach does not necessitate that the quantized weights be confined to any particular range of the form $[0, 2^{b_w})$. Indeed, what controls our approximation accuracy is not the largest possible entry in $\boldsymbol{w}_q$, but rather the number of additions per input element, which is given by $\|\boldsymbol{w}_q\|_1/d$. Therefore, given a budget of $R$ additions per input element, we propose to use a quantization step of $\gamma_w = \|\boldsymbol{w}\|_1/(Rd)$ in (9), so that

$$\mathcal{Q}(w_i) = \text{round}(w_i/\gamma_w). \tag{12}$$

This quantization ensures that the number of additions per input element is indeed as close as possible to the prescribed $R$. We remark that although we assumed unsigned weights, this quantization procedure can also be used for signed weights (after quantization, the positive and negative weights can be treated separately in order to save power, as in Sec. 4).

## 5.2 POWER CONSUMPTION

We emphasize that in PANN, we would not necessarily want to use the same bit width for the activations as in regular quantization. We therefore denote the activation bit width in PANN by $\tilde{b}_x$ to distuingish it from the $b_x$ bits we would use with the regular quantizer. To estimate the power consumed by our approach, note that we have approximately $\|\boldsymbol{w}\|_1$ additions of $\tilde{b}_x$ bit numbers. On

average, each such addition leads to $0.5\tilde{b}_x$ bit flips at the accumulator's output and $0.5\tilde{b}_x$ bit flips in the FF register (see Table 1). The input to the accumulator, however, remains fixed for $\mathcal{Q}_w(w_i)$ times when approximating the $i$th MAC and therefore changes a total of only $d$ times throughout the entire computation in (11), each time with $0.5\tilde{b}_x$ bit flips on average. Thus, overall, the average power per element consumed by PANN is

$$P_{\text{PANN}} = \frac{\|\boldsymbol{w}\|_1 \tilde{b}_x + 0.5\tilde{b}_x d}{d} = (R + 0.5)\tilde{b}_x. \tag{13}$$

This implies that to comply with a prescribed power budget, we can either increase the activation bit width $\tilde{b}_x$ on the expense of the number of additions $R$, or vice versa.

Figure 3(a) depicts the combinations of $\tilde{b}_x$ and $R$ that lead to the same power consumption as that of a $b_x$ bit unsigned MAC, $P_{\text{MAC}}^{\text{u}} = 0.5b_x^2 + 4b_x$ (see (3),(4)), for several values of $b_x$ (different colors). When we traverse such an equal-power curve, we also change the quantization error. Thus, the question is whether there exist points along each curve, which lead to lower errors than those obtained with regular quantization at the bit-width corresponding to that curve.

### 5.3 QUANTIZATION ERROR

Let us compute the quantization error incurred by PANN and compare it to that of a regular uniform quantizer (RUQ). Obviously, the error incurred by the approximation (9) is contributed by both the quantization of the weights and the quantization of the activations. To see how their errors interact, let us assume that $\boldsymbol{w}$ and $\boldsymbol{x}$ are statistically independent random vectors, each with iid components. In this case, if the quantization errors $\boldsymbol{\varepsilon_w} = \boldsymbol{w} - \gamma_w \mathcal{Q}(\boldsymbol{w})$ and $\boldsymbol{\varepsilon_x} = \boldsymbol{x} - \gamma_x \mathcal{Q}(\boldsymbol{x})$ satisfy $\mathbb{E}[\boldsymbol{\varepsilon_w}|\boldsymbol{w}] = 0$ and $\mathbb{E}[\boldsymbol{\varepsilon_x}|\boldsymbol{x}] = 0$, then we can show (see Appendix A.9) that the mean squared error (MSE) between the full precision operation (8) and its quantized version (9) is given by

$$\text{MSE} = \mathbb{E}\left[\left(\boldsymbol{w}^T \boldsymbol{x} - \boldsymbol{w}_q^T \boldsymbol{x}_q\right)^2\right] \approx d\left(\sigma_w^2 \sigma_{\varepsilon_x}^2 + \sigma_x^2 \sigma_{\varepsilon_w}^2\right). \tag{14}$$

Here $\sigma_w^2$, $\sigma_x^2$, $\sigma_{\varepsilon_w}^2$, and $\sigma_{\varepsilon_x}^2$ denote the second-order moments of the elements of $\boldsymbol{w}$, $\boldsymbol{x}$, $\boldsymbol{\varepsilon_w}$, and $\boldsymbol{\varepsilon_x}$, respectively, and the expression on the right results from neglecting second-order terms.

Consider a simplistic scenario where the activations are uniformly distributed in $[0, M_a]$ (recall that activations are non-negative due to the preceding ReLU) and the weights are uniformly distributed in $[-\frac{1}{2}M_w, \frac{1}{2}M_w]$. If we use a RUQ with $b_x$ bits to quantize the activations and a RUQ with $b_w$ bits to quantize the weights, then we have that

$$\sigma_x^2 = \frac{M_x^2}{3}, \qquad \sigma_{\varepsilon_x}^2 = \frac{M_x^2}{12 \cdot 2^{2b_x}}, \qquad \sigma_w^2 = \frac{M_w^2}{12}, \qquad \sigma_{\varepsilon_w}^2 = \frac{M_w^2}{12 \cdot 2^{2b_w}}. \tag{15}$$

This is because the quantization errors are uniformly distributed as $\varepsilon_x \sim U[-2^{-(b_x-1)}, 2^{-(b_x-1)}]$ and $\varepsilon_w \sim U[-2^{-(b_w-1)}, 2^{-(b_w-1)}]$. Substituting (15) into (14), we obtain that the MSE of a RUQ is

$$\text{MSE}_{\text{RUQ}} = \frac{dM_x^2 M_w^2}{12^2}\left(2^{-2b_x} + 4 \cdot 2^{-2b_w}\right). \tag{16}$$

In PANN, we have that $(\varepsilon_w)_i|\boldsymbol{w} \sim U[-\frac{\|\boldsymbol{w}\|_1}{2Rd}, \frac{\|\boldsymbol{w}\|_1}{2Rd}]$, so that $\mathbb{E}[(\varepsilon_w)_i^2|\boldsymbol{w}] = \frac{\|\boldsymbol{w}\|_1^2}{12Rd}$. Therefore,

$$\sigma_{\varepsilon_w}^2 \approx \frac{\mathbb{E}[\|\boldsymbol{w}\|_1^2]}{12(Rd)^2} = \frac{d^2(0.25M_w)^2}{12(Rd)^2} = \frac{M_w^2}{192R^2}, \tag{17}$$

where we used the fact that $|w_i| \sim U[0, 0.5M_w]$. Substituting this expression in (14), we find that using PANN together with a $\tilde{b}_x$ bit RUQ for the activations, we achieve

$$\text{MSE}_{\text{PANN}} = \frac{dM_x^2 M_w^2}{12^2}\left(2^{-2\tilde{b}_x} + \frac{1}{4R^2}\right). \tag{18}$$

To compare between PANN and RUQ, we need to fix a power budget $P$. Given such a budget, (13) dictates that the number of additions in PANN should be set to $R = P/\tilde{b}_x - 0.5$. Substituting this into (19), we obtain that

$$\text{MSE}_{\text{PANN}} = \frac{dM_x^2 M_w^2}{12^2}\left(2^{-2\tilde{b}_x} + \frac{\tilde{b}_x^2}{(2P - \tilde{b}_x)^2}\right). \tag{19}$$

The optimal bit-width for the activations can therefore be found numerically by minimizing $\text{MSE}_{\text{PANN}}$ over $\tilde{b}_x \in \mathbb{Z}^+$. This typically requires evaluating (19) for a small number of candidate bit widths, e.g., $\tilde{b}_x \in \{2, \ldots, 8\}$. See Appendix A.8 for a thorough analysis.

Figure 3(b) shows the ratio between $\text{MSE}_{\text{RUQ}}$ and $\text{MSE}_{\text{PANN}}$ (with the optimal $\tilde{b}_x$) as a function of the bit width of the RUQ, where PANN is tuned to the same power. For the RUQ, we use $b_x = b_w$ as its power consumption (7) is anyway dominated by the larger of them. It can be seen that for low bit widths, PANN has a significant advantage over RUQ (ratio larger than 1). In the Gaussian setting, which is closer to the distribution of DNN weights and activations, the range over which PANN outperforms RUQ is even larger. As we show in Appendix A.8, this behavior is very similar to that observed in deep networks for image classification.

We emphasize that (19) is valid for uniformly distributed weights and activations, which is often not an accurate enough assumption for DNNs. Thus, in practice the best way to determine the optimal bit width $\tilde{b}_x$ is by running the quantized network on a validation set, as summarized in Algorithm 1.

---

**Algorithm 1: Determining the optimal parameters for PANN**

---

1: **Input:** Power budget $P$
2: **Output:** Optimal $\tilde{b}_x, R$
3: **for** each $\tilde{b}_x \in [\tilde{b}_x^{\min}, \tilde{b}_x^{\max}]$ **do**
4:     Set $R = P/\tilde{b}_x - 0.5$ (Eq. (13))
5:     Quantize the weights using Eq. (12) with $\gamma_w = \|\boldsymbol{w}\|/(Rd)$
6:     Quantize the activations to $\tilde{b}_x$ bits using any quantization method
7:     Run the network on a validation set, with multiplications replaced by additions using Eq. (10)
8:     Save the accuracy to $\text{Acc}(\tilde{b}_x)$.
9: **end for**
10: set $\tilde{b}_x \leftarrow \arg\max_{\tilde{b}_x} \text{Acc}(\tilde{b}_x), \quad R \leftarrow P/\tilde{b}_x - 0.5$

---

## 6 EXPERIMENTS

We now examine PANN in DNN classification experiments. We start by examining its performance at post training, and then move on to employ it during training. Here we focus only on the effect of removing the multiplier (vertical arrows in Fig. 1). Namely, we assume all models have already been converted to unsigned arithmetic (recall this by itself reduces a lot of the power consumption).

**PANN at post training** We illustrate PANN's performance in conjunction with a variety of post training quantization methods, including the data free approaches GDFQ (Shoukai et al., 2020) and ZeroQ (Cai et al., 2020), the small calibration set method ACIQ (Banner et al., 2019), and the optimization based approach BRECQ (Li et al., 2021), which is currently the state-of-the-art for post training quantization at low bit widths. Table 2 reports results with ResNet-50 on ImageNet (See Appendix A.5.1 for results with other models). For the baseline methods, we always use equal bit widths for the weights and activations. Each row also shows our PANN variant, which works at the precise same power budget, where we choose the optimal $\tilde{b}_x$ and $R$ using Alg. 1. As can be seen, PANN exhibits only a minor degradation w.r.t. the full-precision model, even when working at the power budget of 2 bit networks. This is while all existing methods completely fail in this regime.

**PANN for quantization aware training** To use PANN during training, we employ a straight-through estimator for backpropagation through the quantizers. Table 3 compares our method to LSQ (Esser et al., 2019), which is a state-of-the-art QAT approach, where in PANN we use LSQ for quantizing the activations. As can be seen, PANN outperforms LSQ for various models and power budgets. In Table 4 we compare our method to the multiplication-free approaches AdderNet (Chen et al., 2020) and ShiftAddNet (You et al., 2020), which are also training-based techniques. For each method, we report the *addition factor*, which is the ratio between its number of additions per layer and a regular layer. For example, AdderNet uses no multiplications but twice as many additions, so that its addition factor is 2. ShiftAddNet, on the other hand, uses one addition and one shift operation. According to You et al. (2020), a shift operation costs between $0.2$ (on FPGA) and $0.8$ (on a 45nm ASIC) an addition operation. Therefore ShiftAddNet's addition factor is between $1.2$ and $1.8$, and for simplicity we regard it as $1.5$. In PANN, we can choose any addition factor $R$, and therefore examine

| POWER (BITS) | | | ACIQ | | ZEROQ | | GDFQ | | BRECQ | |
|---|---|---|---|---|---|---|---|---|---|---|
| | MEM. | LATENCY | BASE. | **OUR** | BASE. | **OUR** | BASE. | **OUR** | BASE. | **OUR** |
| 265 (8) | 1× | 7.5× | 76.02 | 76.10 | 75.90 | 75.77 | 76.17 | 76.05 | 76.10 | 76.05 |
| 217 (6) | 1.3× | 4.7× | 75.41 | 76.05 | 73.57 | 74.65 | 76.05 | 76.02 | 75.86 | 76.01 |
| 134 (5) | 2× | 3.5× | 74.02 | 75.50 | 58.62 | 74.32 | 71.40 | 75.96 | 75.75 | 75.96 |
| 99 (4) | 2.3× | 2.9× | 66.12 | 75.10 | 3.53 | 68.24 | 50.81 | 75.20 | 75.42 | 75.80 |
| 68 (3) | 2× | 2.2× | 7.73 | 74.16 | 1.51 | 68.12 | 0.24 | 74.85 | 68.12 | 74.62 |
| 41 (2) | 3× | 1.1× | 0.20 | 71.55 | 0.10 | 62.96 | 0.13 | 74.32 | 18.80 | 73.21 |

Table 2: **PTQ: Classification accuracy [%] of ResNet-50 on ImageNet (FP 76.11%).** The baselines (Base.) use equal bit widths for weights and activations. This bit width determines the power $P$, reported in first column in units of Giga bit-flips. The power is calculated as $P^{\mathrm{u}}_{\mathrm{mult}} + P^{\mathrm{u}}_{\mathrm{acc}}$ (Eqs. (3),(4)) times the number of MACs in the network. In each row, our variant PANN is tuned to work at the same power budget, for which we choose the optimal $\tilde{b}_x$ and $R$ using Alg. 1.

| Bits (Power), Net | LSQ | PANN |
|---|---|---|
| 18 (2), ResNet-18 | 67.32 | 70.83 |
| 30 (3), ResNet-18 | 69.81 | 71.12 |
| 41 (2), ResNet-50 | 71.36 | 76.65 |
| 68 (3), ResNet-50 | 73.54 | 76.78 |
| 155 (2), VGG-16bn | 71.15 | 73.30 |

| Method | 6/6 | 5/5 | 4/4 | 3/3 |
|---|---|---|---|---|
| Our (1×) | 91.15 | 91.05 | 89.93 | 85.62 |
| Our (1.5×) | 91.52 | 91.50 | 90.05 | 86.12 |
| Our (2×) | 91.63 | 91.61 | 90.10 | 86.84 |
| ShiftAddNet (1.5×) | 87.72 | 87.61 | 86.76 | 85.10 |
| AdderNet (2×) | 67.39 | 65.53 | 64.31 | 63.50 |

Table 3: **QAT: Comparison with LSQ.** Imagenet classification accuracy [%] of various models. We report the bit width of LSQ and power in Giga bit-flips.

Table 4: **QAT: Comparison with multiplier-free methods.** Classification accuracy [%] of ResNet-20 on CIFAR-10. The top row specifies weight/activation bit widths, and the addition factor is specified in parentheses.

our method for $R = 1, 1.5, 2$. We can see in the table that PANN outperforms both AdderNet and ShiftAddNet for all bit widths, even when using a smaller addition factor. Please see more QAT comparisons in Appendix A.5.2

**Runtime memory footprint and latency of PANN**   We now analyze the effect of PANN on other inference aspects besides power. One important aspect is *runtime memory footprint*. When working with batches of image, the runtime memory consumption is dominated by the activations (Mishra et al., 2017) (see discussion on the memory footprint of the weights in Appendix A.7). The optimal number of bits $\tilde{b}_x$ we use for the activations is typically larger than the bit width $b_x$ used in regular quantization. The second column of Table 2 reports the factor $\tilde{b}_x/b_x$ by which the runtime memory of PANN exceeds that of the baseline model. As can be seen, this factor never exceeds 3. In the comparisons with the multiplier-free methods (Table 4), we keep the same bit width for the activations and therefore there is no change in the memory footprint. A second important factor is *latency*. Recall we remove the multiplier and remain only with the accumulator. Since addition is faster than multiplication, one could potentially use a higher clock-rate and thus gain speed. However, if we conservatively assume the original clock-rate, then the latency is increased by $R$ (each multiplication is replaced by $R$ additions). As can be seen in Table 2, the increase in latency is quite small at the lower power budgets. For the multiplier-free methods, we obtain improvement in accuracy even for $R = 1$. In that case, our latency is smaller than that of AdderNet (2×) and ShiftAddNet (1.5×). Please refer to Appendix A.7 for more analyses.

## 7   CONCLUSION

We presented an approach for reducing the power consumption of DNNs. Our technique relies on a detailed analysis of the power consumption of each arithmetic module in the network, and makes use of two key principles: switching to unsigned arithmetic, and employing a new weight quantization method that allows removing the multiplier. Our method substantially improves over existing approaches, both at post-training and when used during training, and allows achieving a significantly higher accuracy at any given power consumption budget.

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

## A APPENDIX

### A.1 POWER SIMULATION ON 5NM PROCESS

Using the Synopsys DesignWare Library[4], we built a Verilog RTL (Register Transfer Logic) simulation which instantiates signed multipliers and signed adders of 2-8 bit widths. For the multipliers, we used a Radix-4 Booth-encoder implementation and for the adders, we used the Ripple Carry implementation. We synthesized these adders and multipliers, with a 5nm cells library at a clock frequency of $1.6GHz$.

In order to analyze the power consumption of each module individually, we used a hierarchical gate-level synthesis where each module is a different utility that does not share logic with any other module. The synthesis result is the gate-level netlist with the logic gates, which is the actual implementation of the multipliers and adders. Then, using Synopsys PrimeTime PX[5] (PTPX), which accurately reflects ASIC power consumption, we ran a simulation with uniformly distributed random inputs on the gate-level netlist and measured the power of multiplication and addition instructions.

It should be noted that, as opposed to papers experimenting with FPGAs, our primary interest is in modern integrated chips (ASICs), like CPUs and GPUs. Gate-level simulations of the type we use here are the *only* practical (and most accurate) way to estimate ASIC power consumption. This is because even if we had fabricated this netlist as part of a real ASIC (which would cost millions of dollars), it would still be impossible to measure the power of a small portion of the chip accurately.

---

[4]https://www.synopsys.com/silicon-design.html
[5]https://www.synopsys.com/support/training/signoff/primetimepx-fcd.html

In figures 4(a) and 4(b) we depict the average power consumed by a multiplication of two $b$-bit numbers and by an addition of two $b$-bit numbers, respectively. We can see that these power measurements agree with our Python simulation, which we discuss in the next section. A remark is in place about the slight deviation between the 5nm simulation and our Python simulation seen in Fig. 4(a). This deviation implies that the advantage of PANN over a regular DNN is slightly more substantial than that reported in the paper. Specifically, since our theoretical model slightly underestimates the power consumed by a multiplier at high bit-widths, we actually underestimate the benefit of our method, which avoids multiplications. The configuration files will be published online to enable easy reproduction of the results.

In Table 5, we report the amount in [%] of the dynamic power and the static power as measured using our 5nm simulation.

| MEASURED | 2-BIT | 3-BIT | 4-BIT | 5-BIT | 6-BIT | 7-BIT | 8-BIT | 32-BIT |
|---|---|---|---|---|---|---|---|---|
| DYNAMIC POWER (MULTIPLIER) | 59 | 57 | 55 | 51 | 50 | 51 | 51 | – |
| STATIC POWER (MULTIPLIER) | 41 | 43 | 45 | 49 | 50 | 49 | 49 | – |
| DYNAMIC POWER (ADDER) | 61 | 60 | 59 | 58 | 58 | 55 | 56 | 60 |
| STATIC POWER (ADDER) | 39 | 40 | 41 | 42 | 42 | 45 | 44 | 40 |

Table 5: **Static power vs. dynamic power in [%].** We can see that overall the dynamic power constitutes a major portion of the total power.

### A.2    POWER SIMULATION IN PYTHON

Using Python, we implemented a simple serial adder and two types of multipliers: a simple serial multiplier and a Radix-2 Booth encoding multiplier. A serial multiplier follows the long multiplication concept in which each bit of the multiplicand multiplies the multiplier word. This results in a word of length $b + 1$ bits at most, called a partial product. Going over all bits in the multiplicand results in $b$ partial products that need to be summed. The Booth encoder is more efficient in terms of the number of partial products that need to be summed. It comprises an encoder that follows a lookup table and directs whether to perform shift, addition or subtraction, basing on consecutive pairs of bits of the multiplicand. For example, suppose we want to multiply a number $x$ by 15, which is 1111 in binary representation. The serial multiplier performs the computation $x \times \left(2^3 + 2^2 + 2^1 + 2^0\right)$ while the Booth encoder multiplier computes $x \times \left(2^4 - 2^0\right)$, and saves two sums. Those partial products are summed by half and full adders that are the major area and power consumers of a multiplier.

We are focusing on the dynamic power, which is a prominent source of power consumption and linearly depends on the switching activity. Therefore, in order to estimate the power, we measured the average number of bit toggles per instruction (*e.g.*, multiplication and addition). We counted the toggles at the inputs of each half or full 1-bit adder component, both for the $b$-bit multiplier and for the $b$-bit adder. Please see an illustrative example in Fig. 6. In figures 7,8 we show the average number of toggles per instruction that were counted in the multiplier and in the adder using signed and unsigned numbers, respectively. We ran our simulation with data drawn from a uniform distribution and a Gaussian one. We took the uniform distribution to be over the range $[-2^{b-1}, 2^{b-1})$. As for the Gaussian, we first drew $N$ full precision numbers from $N(0,1)$. Then, we divided them by their maximum (in absolute value), multiplied by $2^{b-1}$, and rounded to the closest integer. We clipped the values to the range $[-2^{b-1}, 2^{b-1})$ in order to eliminate outliers (specifically the number $2^{b-1}$). In all our experiments, we took $N = 36000$. Please see an example histogram for the Gaussian distributed numbers in Fig. 5(b), where $b = 8$.

### A.3    OBSERVATION-1

When working with an accumulator having a large bit width $B$ (*e.g.* $B = 32$), a dominant source of power consumption is the bit toggling in its inputs, which is $0.5B$ per instruction on average. This comes from the 2's complement representation. Hence, a significant amount of power can be saved when switching to unsigned numbers. For example, in the right plot of Fig. 8 we show that the power in the accumulator inputs is reduced from 16 (assuming a 32-bit accumulator) to $0.5b_{acc}$ where $b_{acc}$ is the bit width at the input to the accumulator.

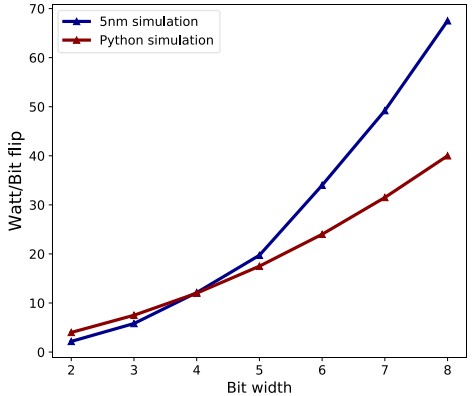

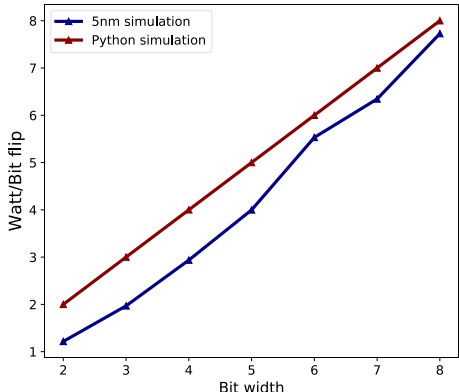

(a) Measuring the power of a multiplication operation.   (b) Measuring power of an addition operation.

Figure 4: (a) In red we plot the power consumed by a multiplication operation as measured in our Python simulation. This curve agrees with the theoretical power model in Eq. (3) in the paper, i.e. $P_{\text{mult}} = 0.5b^2 + b$ (see left side of Fig. 7). In blue, we plot the power measurements on a 5-nm silicon process. In order to ignore the different power units and set both measurements on the same axis, we scaled the results of the 5nm power simulation so that the curves intersect at bit width of $4$. (b) In this experiment, we measured the power consumed by a $b$-bit accumulator without the FF (thus, here $b_{acc} = b$). In red we can observe the power measured in our Python simulation, which is very close to our theoretical model $P_{\text{acc}} = 0.5b + 0.5b = b$ (see right side of Fig. 7). We can see that the power measurements for the 5nm silicon process (blue) nicely agree with our Python simulation. Here we scaled the results of the 5nm simulation using the same factor we found in Fig. 4(a).

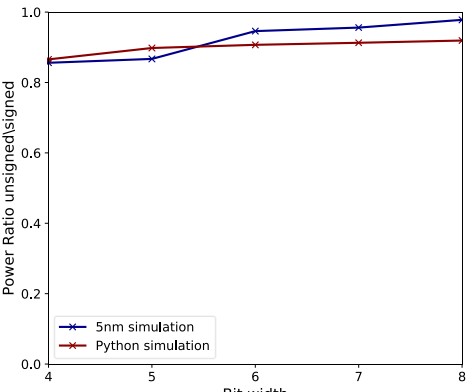

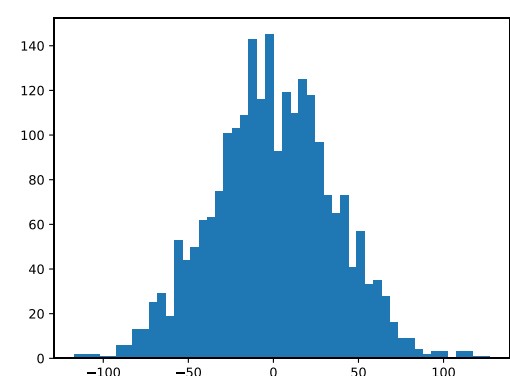

(a) Switching to unsigned numbers does not affect the multiplier.   (b) An example quantized Gaussian distribution.

Figure 5: (a) Using our Python simulation, We measured the average number of toggles between unsigned multiplication and signed multiplication for bit widths of $4$-$8$. As can be seen, we obtained an average ratio of $92\%$ (red curve). This observation is aligned with the power measured in our 5nm silicon process (blue curve). (b) Unlike the uniform distribution, we can see that the majority of values occupies roughly half of the allowed interval and therefore on average, we observe a bit less toggles than with the uniform distribution (here the bit width is $b = 8$).

As for the multiplier, switching to unsigned values turns out to have a negligible effect in terms of power consumption (left plot of Fig. 8). In Fig. 5(a) we show the ratio between the power consumed by multiplication of unsigned numbers and multiplication of signed numbers, as measured in our Python simulation and in the 5nm silicon process. As can be seen, this ratio is close to $1$ for all bit

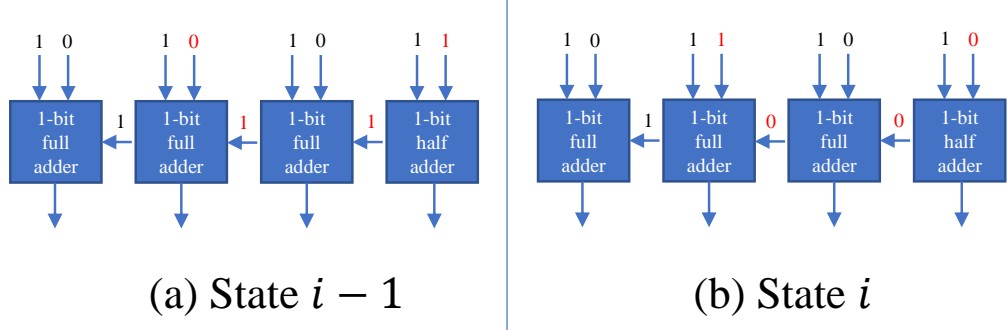

(a) State $i - 1$        (b) State $i$

Figure 6: **Counting bit toggles in the multiplier's internal adders.** We depict a snapshot of the multiplier's internal components at two consecutive addition instructions. At state $i - 1$ we sum $1111$ and $0001$ and at state $i$ we sum $1111$ and $0100$. In our python simulation, we compare between the bit status of consecutive operations therefore in this example, we will count four toggles (two in the input words and two in the internal carry outputs).

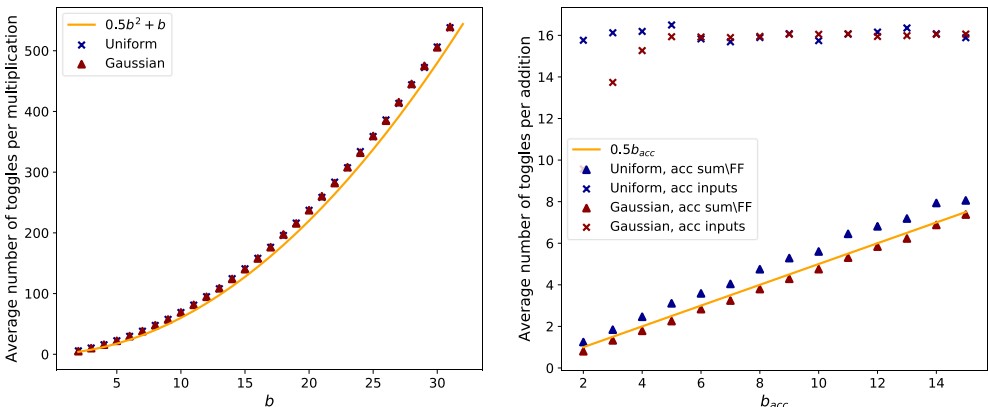

Figure 7: **Python simulation for signed integers.** On the left, we plot the power consumed by the multiplier. We counted the toggles at the inputs of the multiplier (row 1 in Table 1 of the paper) as well as the toggles inside its internal units (row 2 in Table 1). We can see that the power measured in our simulations closely agrees with power model in Eq. (1), $0.5b^2 + b$. On the right, we plot the power consumed by the accumulator, where the label "acc inputs" refers to the power due to the bit flips at its input (row 3 in Table 1). In this case $B = 32$ and therefore we observe a constant power of $16$. The label "acc sum" refers to the power consumed due to the toggles at the output of the accumulator (row 4 in Table 1) which also toggles the bits in the FF. Again, the simulation agrees with our model.

widths. Therefore, we adopt the same power model for the unsigned multiplier case as in the signed setting.

### A.3.1 SWITCHING TO UNSIGNED ARITHMETIC

Figure 11(a) compares the average power consumption of a signed MAC to that of an unsigned MAC for a 32 bit accumulator. Specifically, we are dividing $P_{\text{mult}}^{\text{u}} + P_{\text{acc}}^{\text{u}}$ by $P_{\text{mult}} + P_{\text{acc}}$. In this setting, it can be seen for example that when working with $b = 4$ bits for the weights and activations, unsigned MACs are 33% cheaper in power. The approach we suggest for switching a linear layer (*e.g.*, convolution, fully connected) to work with unsigned arithmetic is illustrated schematically in Fig. 11(b).

**Accumulator bit width**  The bit width of the accumulator is commonly chosen to be 32. One of the main reasons for that is this allows flexibility in changing the bit widths of the activations and weights

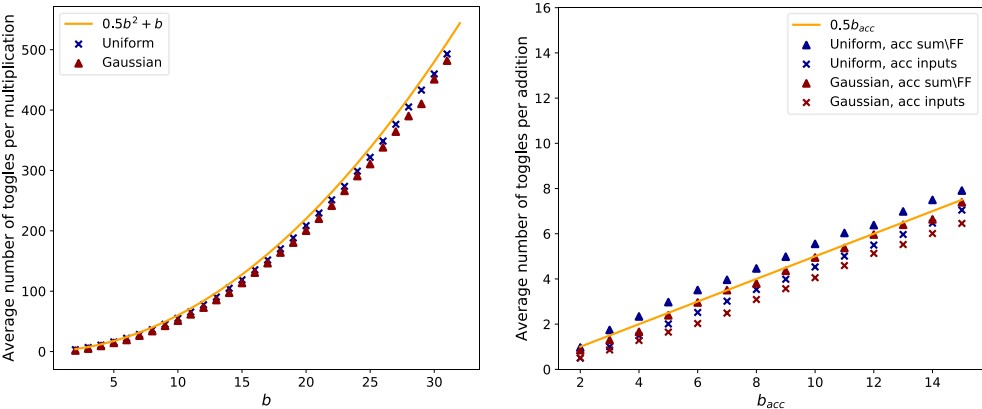

Figure 8: **Python simulation for unsigned integers.** Here we repeat the experiment of Fig. 7, but with numbers drawn only from the interval $[0, 2^{b-1})$ for both the uniform and the Gaussian distributions. On the left, we can see that the overall power of the multiplier has not changed much and is aligned with Eq. (3) in the paper. However, on the right we can see that due to the use of unsigned values, the power consumed by the toggling at the accumulator inputs is dramatically reduced ($0.5b_{\mathrm{acc}}$ instead of $0.5B$). The rest of the power contributors did not change much.

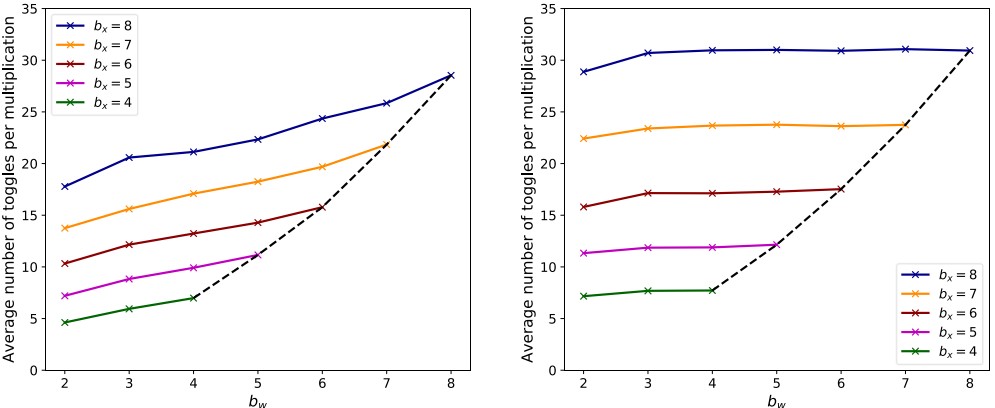

Figure 9: **Working with $b_w < b_x$ in a Booth encoder multiplier**. On the right, we uniformly drew $b_x$-bit numbers from $[-2^{b_x-1}, 2^{b_x-1})$ and $b_w$-bit numbers from $[-2^{b_w-1}, 2^{b_w-1})$, for various $b_w \leq b_x$. We can see that the power is affected only by the larger bit width ($b_x$), and remains nearly constant when reducing only $b_w$. On the left, we repeat the same experiment, however with unsigned values, where the $b_x$-bit input is uniformly drawn from $[0, 2^{b_x-1})$ and the $b_w$-bit input from $[0, 2^{b_w-1})$. Here, there is a slight benefit in decreasing only $b_w$. The black dashed curve connects the power measurements for the cases where $b_w = b_x$, and follows the parabolic behaviour.

(*e.g.*, from 4-bit to 8-bit and vice versa). Nevertheless, if we are not concerned with flexibility, then when quantizing the activations and weights to less than 8 bits, we can use an accumulator with less than 32 bits. The required accumulator bit width $B$ can be calculated by

$$B = b_x + b_w + 1 + \log\left(k^2 C_{\mathrm{in}}\right), \tag{20}$$

where $k$ is the convolution kernel size, and $C_{\mathrm{in}}$ is the number of input channels. In Table 6 we analyze the case of ResNet networks. We choose the layer with the largest value of $k^2 C_{\mathrm{in}}$, which is 3x3x512 (Table 1 in He et al. (2016)). We calculate the required bit width for the accumulator (*e.g.*, $B$) when the activations and weights are quantized to 2-6 bits. In addition, we calculate the power save in [%] when switching to unsigned arithmetic. As can be seen, even with smaller accumulator bit widths, switching to unsigned arithmetic leads to a significant saving in power. This is also visually

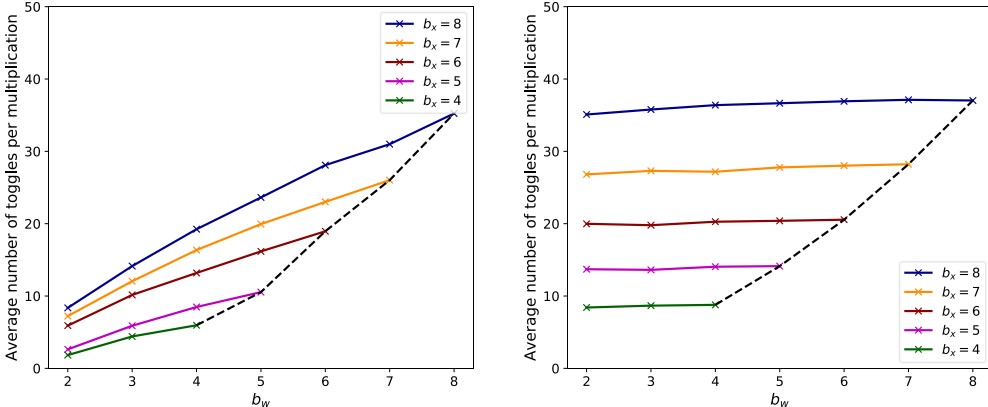

Figure 10: **Working with $b_w < b_x$ in a simple serial multiplier**. On the right, we uniformly drew $b_x$-bit numbers from $[-2^{b_x-1}, 2^{b_x-1})$ and $b_w$-bit numbers from $[-2^{b_w-1}, 2^{b_w-1})$ such that $b_w \le b_x$. We can see that the power is affected by the larger bit width ($b_x$). On the left, we repeat the experiment however with unsigned values, where the $b_x$-bit input is uniformly drawn from $[0, 2^{b_x-1})$ and the $b_w$-bit input from $[0, 2^{b_w-1})$. Here, there is more benefit in decreasing only $b_w$. The black dashed curve connect the power measurements for the cases where $b_w = b_x$, and follows the parabolic behaviour.

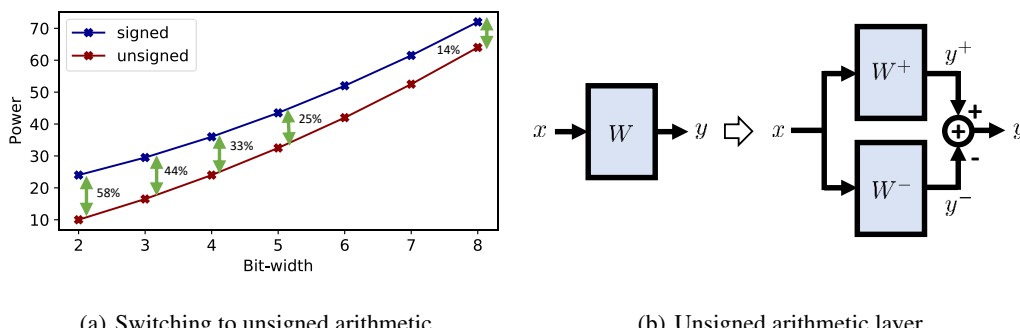

(a) Switching to unsigned arithmetic                    (b) Unsigned arithmetic layer

Figure 11: (a) Based on our power model, we show that a significant amount of power can be saved with minimal effort, by switching to work with unsigned numbers. Here we assume a 32 bit accumulator. (b) Any weight matrix $W$ can be split into its positive and negative parts (See Sec. 4). Assuming the elements of its input $x$ are non-negative (due to the preceding ReLU), this makes all MACs unsigned and thus substantially reduces power consumption.

illustrated in Fig. 12, where we repeat the experiment of Fig. 1 bit with a 17 bit accumulator for the 2-bit networks, and with a 21 bit accumulator for the 4-bit networks.

### A.4    OBSERVATION-2

We now analyze the case where the inputs of the multiplier have different bit widths, $b_x$ and $b_w$. In Fig. 9 we show the average bit toggles in signed and unsigned Booth encoder multiplication, for $b_w \le b_x$. In Fig. 10 we show the same analysis for the simple serial multiplier. We observe that when working with signed numbers (the common setting), the power is mostly affected by the larger bit width ($b_x$ in this case).

In the case of unsigned numbers, there is some power save when reducing one of the bit widths. In other words, Eq. (7) in the paper is accurate for the popular signed case and behaves as an upper bound for the unsigned case. The difference between Eq. (7) and the actual power consumption in the unsigned setting is more dominant for the simple serial multiplier. Therefore, in certain settings,

|  | 2-BIT | 3-BIT | 4-BIT | 5-BIT | 6-BIT |
|---|---|---|---|---|---|
| REQUIRED BIT WIDTH $B$ | 17 | 19 | 21 | 23 | 25 |
| POWER SAVE FOR A $B$ BIT ACCUMULATOR | 39% | 28% | 21% | 16% | 13% |
| POWER SAVE FOR A 32 BIT ACCUMULATOR | 58% | 44% | 33% | 25% | 19% |

Table 6: **Required accumulator bit width.** Here we compute the bit width required for the accumulator, according the largest linear layer in ResNets (which is 3x3x512). For example, in case of 2 bit width activations and weights, we might use an accumulator with 16 bits and not 32 bits. In the last two rows we report the power save in [%] when switching to unsigned arithmetic. Just like in the 32 bit accumulator case, when working with lower bit width accumulators, we can obtain a significant reduction in power by switching to unsigned arithmetic.

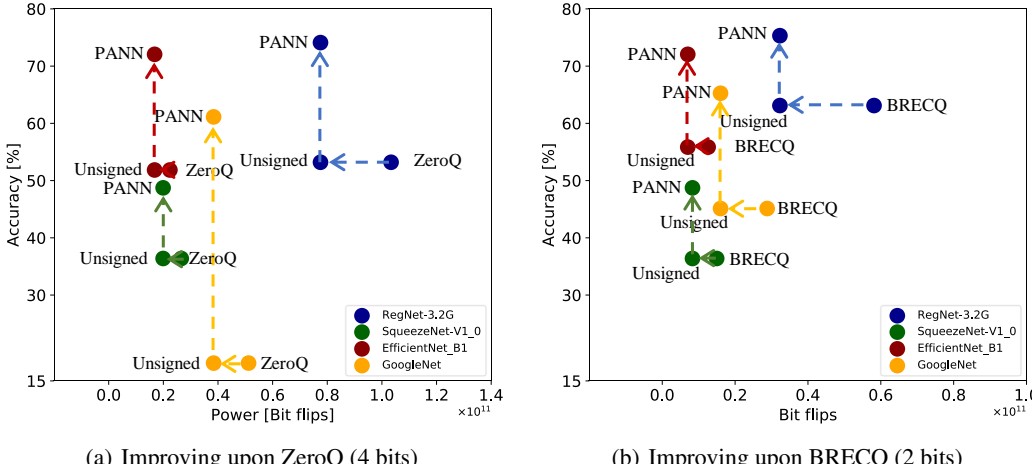

(a) Improving upon ZeroQ (4 bits)  (b) Improving upon BRECQ (2 bits)

Figure 12: **Power-accuracy trade-off at post training with different bit width accumulators**. We repeat the experiment of Fig. 1, however this time we assume a 21 bits accumulator in (a) and a 17 bit accumulator in (b). Theretofore, when converting the quantized models to work with unsigned arithmetic ($\leftarrow$), it cuts down 21% of the power consumption in (a) and 39% in (b).

there is an additional benefit of switching to unsigned arithmetic, which we did not report in the experiments in the paper (i.e. the horizontal arrows in Fig. 1 should actually be slightly longer in some cases). Yet, this effect is relatively small compared to the reduction in bit flips in the accumulator.

We validated our observation on the 5-nm silicon process setup. We used $8 \times 8$ multiplier and measured the power when one of the inputs was drawn uniformly from $[0, 2^7)$ and the other from $[0, 2^3)$. We got 95% of the power that was measured when both inputs were drawn from $[0, 2^7)$. In case of signed values, when one of the inputs was drawn uniformly from $[-2^7, 2^7)$ and the other from $[-2^3, 2^3)$ we observed 100% of the power that was measured when both inputs were drawn from $[-2^7, 2^7)$.

Note that in order to avoid changing the multiplier's architecture, when switching to unsigned numbers, we use only half the range allowed by the bit width $b_x$, i.e. $[0, 2^{b_x-1})$. Therefore, we obtain a representation with half of the $2^{b_x}$ levels of the signed case (note that we also need to replace $-2^{b_x-1}$ with $-2^{b_x-1}+1$). If we permit architectural changes, then a better way to represent unsigned numbers would be to replace the signed multiplier by a $(b_x+1) \times (b_x+1)$ multiplier that can support unsigned and signed multiplications but consumes much more power, or to work with a $b_x \times b_x$ unsigned multiplier that allow representation in the full interval of $[0, 2^{b_x})$ and then will follow Eq. (7) again.

## A.5 Additional results

### A.5.1 Post training quantization

Similarly to Table 2, we now examine PANN's performance on additional networks. We use the same methods for activations quantization as in Table 2: ACIQ[6] (Banner et al., 2019), ZeroQ[7] (Cai et al., 2020), GDFQ[8] (Shoukai et al., 2020) and BRECQ[9] (Li et al., 2021). We also add Dynamic Quantization, which quantizes the activation and weights on the fly at inference time, according to their dynamic ranges. The results are reported in Tables 7-9.

We first change all networks to work with unsigned arithmetic and measure the classification accuracy, which serves as a baseline (see left side of each method 'Base.'). As mentioned, this step already saves a significant amount of power, without any change in classification accuracy, compared to signed MAC arithmetic. We use the unsigned MAC power consumption as the power budget $P$ and follow Alg. 1. We report the classification accuracy on the right side of each columns (see 'Our').

| Power (Bits) | Dynamic | | ACIQ | | ZeroQ | | GDFQ | | BRECQ | |
|---|---|---|---|---|---|---|---|---|---|---|
| | Base. | **Our** | Base. | **Our** | Base. | **Our** | Base. | **Our** | Base. | **Our** |
| 116 (8) | 69.77 | 69.78 | 69.61 | 69.67 | 69.67 | 69.68 | 69.75 | 69.71 | 69.73 | 69.72 |
| 95 (6) | 66.56 | 69.50 | 69.05 | 69.60 | 67.51 | 69.55 | 69.20 | 69.35 | 69.70 | 69.71 |
| 76 (5) | 55.52 | 69.12 | 67.18 | 69.53 | 54.76 | 69.50 | 68.60 | 69.12 | 69.50 | 69.56 |
| 43 (4) | 0.33 | 68.88 | 55.00 | 69.26 | 26.50 | 69.10 | 60.61 | 69.01 | 68.69 | 68.29 |
| 30 (3) | 0.11 | 68.28 | 1.50 | 68.43 | 0.23 | 68.20 | 19.88 | 68.52 | 65.20 | 67.34 |
| 18 (2) | 0.09 | 63.62 | 0.11 | 66.68 | 0.10 | 66.12 | 0.12 | 68.11 | 43.67 | 66.73 |

Table 7: **Classification accuracy [%] of ResNet-18 on ImageNet (FP: 69.77%).** The baselines (Base.) use equal bit widths for weights and activations (leftmost column). The bit width determines the power $P$, which we specify in units of Giga bit-flips. The power is calculated as $P_{\text{mult}}^{\text{u}} + P_{\text{acc}}^{\text{u}}$ (Eqs. (3),(4)) times the number of MACs in the network. ($1.82 \times 10^9$ in ResNet-18). In each row, our variant is tuned to work at the same power budget, for which we choose the optimal $\tilde{b}_x$ and $R$ using Alg. 1.

| Power (Bits) | Dynamic | | ACIQ | | ZeroQ | | GDFQ | | BRECQ | |
|---|---|---|---|---|---|---|---|---|---|---|
| | Base. | **Our** | Base. | **Our** | Base. | **Our** | Base. | **Our** | Base. | **Our** |
| 21 (8) | 71.82 | 71.79 | 69.73 | 69.71 | 71.79 | 71.53 | 71.88 | 71.76 | 71.95 | 71.85 |
| 19 (6) | 62.13 | 64.13 | 66.16 | 67.18 | 69.35 | 69.58 | 70.48 | 70.52 | 71.36 | 71.55 |
| 11 (5) | 13.11 | 59.55 | 27.06 | 61.14 | 60.49 | 64.33 | 65.32 | 68.31 | 70.30 | 70.98 |
| 8 (4) | 3.56 | 51.25 | 2.32 | 55.13 | 13.92 | 62.14 | 50.96 | 66.02 | 65.12 | 69.12 |
| 5 (3) | 0.05 | 49.33 | 0.09 | 50.23 | 0.06 | 61.11 | 31.19 | 64.13 | 55.14 | 67.85 |
| 3 (2) | 0.01 | 23.26 | 0.07 | 35.55 | 0.03 | 48.12 | 1.55 | 51.12 | 25.91 | 61.08 |

Table 8: **Classification accuracy [%] of Mobilenet-V2 on ImageNet (FP: 71.91%).** The baselines (Base.) use equal bit widths for weights and activations (leftmost column). The bit width determines the power $P$, which we specify in units of Giga bit-flips. The power is calculated as $P_{\text{mult}}^{\text{u}} + P_{\text{acc}}^{\text{u}}$ (Eqs. (3),(4)) times the number of MACs in the network. ($0.33 \times 10^9$ in MobileNet-V2). In each row, our variant is tuned to work at the same power budget, for which we choose the optimal $\tilde{b}_x$ and $R$ using Alg. 1.

In figures 13-14, we demonstrate PANN at post training for different networks under power constrains of 4-bit and 2-bit unsigned MAC. In each figure, we start by running the specified approach to

---

[6]https://github.com/submission2019/cnn-quantization

[7]https://github.com/amirgholami/ZeroQ

[8]https://github.com/xushoukai/GDFQ

[9]https://github.com/yhhhli/BRECQ

| POWER (BITS) | DYNAMIC | | ACIQ | | ZEROQ | | GDFQ | | BRECQ | |
|---|---|---|---|---|---|---|---|---|---|---|
| | BASE. | **OUR** | BASE. | **OUR** | BASE. | **OUR** | BASE. | **OUR** | BASE. | **OUR** |
| 994 (8) | 73.28 | 73.31 | 73.24 | 73.13 | 73.29 | 73.30 | 73.34 | 73.25 | - | - |
| 652 (6) | 72.15 | 73.23 | 73.02 | 73.05 | 73.18 | 73.12 | 73.31 | 73.15 | - | - |
| 505 (5) | 64.05 | 72.88 | 72.31 | 73.02 | 71.11 | 71.92 | 72.25 | 73.02 | - | - |
| 373 (4) | 51.13 | 72.06 | 66.20 | 72.03 | 64.19 | 70.19 | 67.05 | 71.66 | - | - |
| 256 (3) | 2.15 | 70.55 | 31.22 | 71.18 | 20.88 | 69.95 | 51.16 | 71.12 | - | - |
| 155 (2) | 0.56 | 69.95 | 0.13 | 71.02 | 0.18 | 66.62 | 3.63 | 67.96 | - | - |

Table 9: **Classification accuracy [%] of VGG-16bn on ImageNet (FP: 73.35%).** The baselines (Base.) use equal bit widths for weights and activations (leftmost column). The bit width determines the power $P$, which we specify in parentheses in units of Giga bit-flips. The power is calculated as $P_{mult}^u + P_{acc}^u$ (Eqs. (3),(4)) times the number of MACs in the network. ($15.53 \times 10^9$ in VGG-16bn). In each row, our variant is tuned to work at the same power budget, for which we choose the optimal $\tilde{b}_x$ and $R$ using Alg. 1. We failed to run BRECQ due to a CUDA 'out of memory' error.

quantize the bits and the activation to 4 or 2 bits (see caption). We measure the power in bit flips. Specifically, the power of each signed MAC is calculated by $P_{mult} + P_{acc}$ (Eqs. (1),(2)) times the number of MACs in the network. Then, we switch to unsigned arithmetic ($\leftarrow$). In this case the power is calculated by $P_{mult}^u + P_{acc}^u$ (Eqs. (3),(4)) times the number of MACs in the network. For simplicity, let us denote the total power as $P^u$. As can be seen, this stage save power but does not change the accuracy (points marked as 'Unsigned'). Constraining to the same power budget $P^u$, we apply PANN ($\uparrow$). Using Alg. 1, we calculate the optimal activation bit width and the corresponding addition factor (points marked as 'PANN'). We can see a dramatic improvement in classification accuracy without any change in the power consumption.

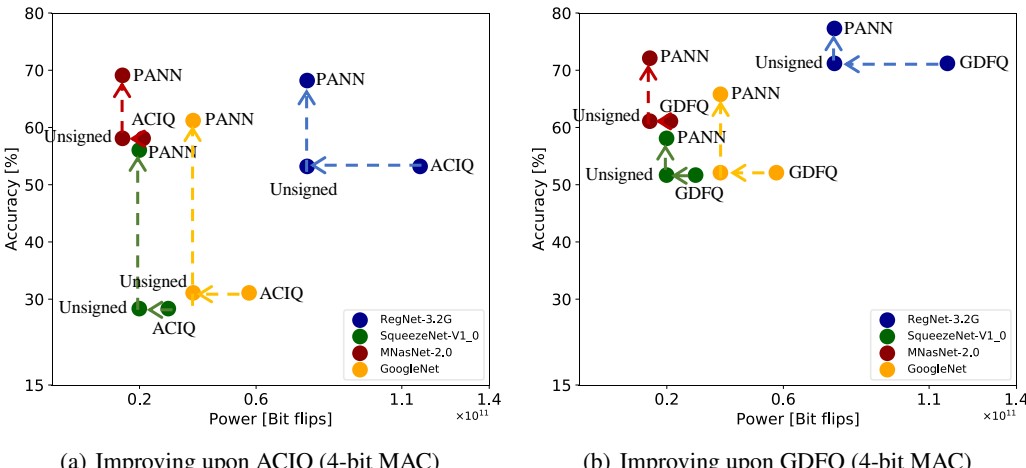

(a) Improving upon ACIQ (4-bit MAC)        (b) Improving upon GDFQ (4-bit MAC)

Figure 13: **Power-accuracy trade-off at post training**. For each pre-trained full-precision model, we used (a) ACIQ (Banner et al., 2019) and (b) GDFQ (Shoukai et al., 2020) to quantize the weights and activations to 4 bits at post-training. Converting the quantized models to work with unsigned arithmetic ($\leftarrow$), already cuts down 33% of the power consumption (assuming a 32 bit accumulator). Using our PANN approach to quantize the weights (at post-training) and remove the multiplier ($\uparrow$), further improves model accuracy for the same power level.

### A.5.2 QUANTIZATION AWARE TRAINING

In Table 10, we report the classification accuracy of different networks on ImageNet, when we use PANN during training. Each row defines a specific power budget, corresponding to 2-bit, 3-bit and

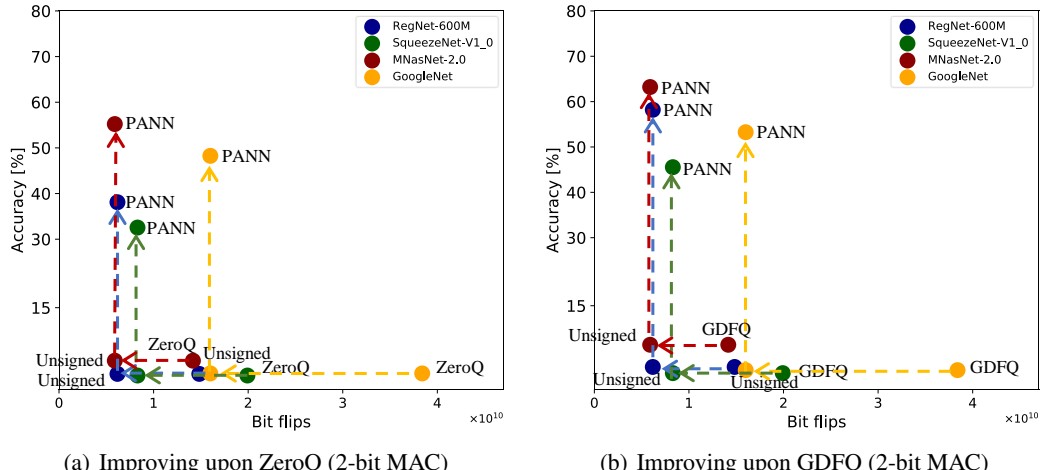

(a) Improving upon ZeroQ (2-bit MAC)      (b) Improving upon GDFQ (2-bit MAC)

Figure 14: **Power-accuracy trade-off at post training**. For each pre-trained full-precision model, we used (a) ZeroQ (Cai et al., 2020) and (b) GDFQ (Shoukai et al., 2020) to quantize the weights and activations to 2 bits at post-training. Converting the quantized models to work with unsigned arithmetic ($\leftarrow$), already cuts down 58% of the power consumption (assuming a 32 bit accumulator). Using our PANN approach to quantize the weights (at post-training) and remove the multiplier ($\uparrow$), further improves model accuracy for the same power level.

4-bit unsigned MACs (weights and activations have equal bit widths). We compare our results to LSQ (Esser et al., 2019), whose accuracy is reported in parentheses. Note that the total number of bit flips differs between the networks, because each has a different number of MACS in its forward pass. Therefore, instead of specifying results as a function of the total number of bit flips, we report results as a function of the bit-width. Each bit-width defines a power budget for which we tune PANN (Alg. 1), where we use LSQ to quantize the activations. We can see that PANN outperforms LSQ at all power budgets.

| Power (bit-width) | ResNet-18 | ResNet-34 | ResNet-50 | ResNet-101 | VGG-16bn |
|---|---|---|---|---|---|
| FP | 70.13 | 73.88 | 76.87 | 77.55 | 73.33 |
| 2 | 70.03 (67.32) | 72.54 (71.21) | 76.65 (71.36) | 77.13 (75.21) | 73.30 (71.15) |
| 3 | 70.12 (69.81) | 73.87 (72.88) | 76.78 (73.54) | 77.24 (76.62) | 73.31 (73.26) |
| 4 | 70.10 (70.13) | 73.96 (73.90) | 76.81 (76.89) | 77.33 (77.52) | 73.46 (73.51) |

Table 10: **PANN for QAT**. Here we report more results of PANN for quantization aware training. In parentheses we report the classification accuracy [%] of LSQ on Imagenet, where both activations and weights are quantized to 2,3 or 4 bits. As for PANN, we follow Alg. 1 to calculate the optimal activation bit width and addition factor.

### A.5.3 ADDITIONAL COMPARISONS TO MULTIPLICATION-FREE METHODS

In tables 11-12 we report additional comparisons with the recent multiplication free methods ShiftAddNet (You et al., 2020) and AdderNet (Chen et al., 2020), this time on the CIFAR100 and MHEALTH Banos et al. (2014) datasets. Here again PANN is used during training, like the competing methods.

### A.6 HYPER PARAMETERS FOR PANN IN QAT

### A.6.1 LSQ

In all experiments we used the SGD optimizer with momentum of 0.9 and weight decay of $10^{-4}$. We used the softmax cross entropy loss. Unlike the original paper, we started the training from

| METHOD | 6/6 | 5/5 | 4/4 | 3/3 |
|---|---|---|---|---|
| OUR (1×) | 66.16 | 64.50 | 62.80 | 55.51 |
| OUR (1.5×) | 66.23 | 65.8 | 63.58 | 56.85 |
| OUR (2×) | 66.90 | 66.50 | 63.99 | 57.51 |
| SHIFTADDNET (1.5×) | 64.08 | 64.05 | 63.23 | 61.31 |
| ADDERNET (2×) | 41.57 | 35.20 | 29.19 | 21.50 |

Table 11: **QAT: Comparison with multiplier-free methods.** Classification accuracy [%] of multiplier-free methods on CIFAR-100. The top row specifies weight/activation bit widths, and the addition factor is specified in parentheses.

| METHOD | 6/6 | 5/5 | 4/4 | 3/3 |
|---|---|---|---|---|
| OUR (1×) | 95.01 | 84.13 | 65.36 | 59.9 |
| OUR (1.5×) | 95.05 | 85.91 | 68.96 | 62.32 |
| OUR (2×) | 95.34 | 87.36 | 70.82 | 62.51 |
| SHIFTADDNET YOU ET AL. (2020) (1.5×) | 85.61 | 63.34 | 35.77 | 18.19 |
| ADDERNET CHEN ET AL. (2020) (2×) | 89.31 | 68.21 | 26.77 | 10.56 |

Table 12: **QAT: Comparison with multiplier-free methods.** Classification accuracy [%] of multiplier-free methods on the MHEALTH dataset Banos et al. (2014). The top row specifies weight/activation bit widths, and the addition factor is specified in parentheses.

pre-trained networks and with a smaller initial learning rate of $10^{-3}$. Please refer to Table 13 for PANN details, scheduling and number of epochs used for the training.

| ARCH. | QAT ($b_x/b_w$) | P | $\tilde{b}_x$ | R | LR SCHEDULE | $B_s$ | EPOCHS |
|---|---|---|---|---|---|---|---|
| RESNET-18 | LSQ (2/2) | 18 | 3 | 2.83 | ×0.1 EVERY 25 EPOCHS | 128 | 75 |
| RESNET-18 | LSQ (3/3) | 30 | 6 | 2.5 | ×0.1 EVERY 25 EPOCHS | 128 | 75 |
| RESNET-18 | LSQ (4/4) | 43 | 6 | 3.5 | ×0.1 EVERY 25 EPOCHS | 128 | 75 |
| RESNET-34 | LSQ (2/2) | 36 | 3 | 2.83 | ×0.1 EVERY 20 EPOCHS | 64 | 60 |
| RESNET-34 | LSQ (3/3) | 61 | 6 | 2.5 | ×0.1 EVERY 20 EPOCHS | 64 | 60 |
| RESNET-34 | LSQ (4/4) | 88 | 6 | 3.5 | ×0.1 EVERY 20 EPOCHS | 64 | 60 |
| RESNET-50 | LSQ (2/2) | 41 | 3 | 2.83 | ×0.1 EVERY 20 EPOCHS | 64 | 60 |
| RESNET-50 | LSQ (3/3) | 68 | 6 | 2.5 | ×0.1 EVERY 20 EPOCHS | 64 | 60 |
| RESNET-50 | LSQ (4/4) | 99 | 6 | 3.5 | ×0.1 EVERY 20 EPOCHS | 64 | 60 |
| RESNET-101 | LSQ (2/2) | 78 | 3 | 2.83 | ×0.1 EVERY 20 EPOCHS | 64 | 60 |
| RESNET-101 | LSQ (3/3) | 128 | 6 | 2.5 | ×0.1 EVERY 20 EPOCHS | 64 | 60 |
| RESNET-101 | LSQ (4/4) | 187 | 6 | 3.5 | ×0.1 EVERY 20 EPOCHS | 64 | 60 |
| VGG-16BN | LSQ (2/2) | 155 | 3 | 2.83 | ×0.1 EVERY 20 EPOCHS | 64 | 60 |
| VGG-16BN | LSQ (3/3) | 279 | 6 | 2.5 | ×0.1 EVERY 20 EPOCHS | 64 | 60 |
| VGG-16BN | LSQ (4/4) | 372 | 6 | 3.5 | ×0.1 EVERY 20 EPOCHS | 64 | 60 |

Table 13: **Hyper-parameters used in LSQ.** When using pure LSQ as the baseline approach, we quantize both the weights and the activations to the same bit width as specified in the second column ($b_x/b_w$). Then, when applying PANN, we keep the exact training regime and the quantized activations, and only change the quantized weights to be calculated by PANN. Here we report the optimal bit width for the activations and the corresponding addition factor (Alg. 1).

### A.6.2 MULTIPLIER FREE APPROACHES

In all experiments we followed the training regime described in (You et al., 2020). Specifically, for CIFAR10 or CIFAR100 we used a batch size of 256, and 160 epochs. The initial learning rate was 0.1 and then divided by 10 at the 80-th and the 120-th epoch. We used the SGD optimizer with momentum of 0.9 and weight decay of $1e - 4$. For the MHEALTH dataset, we used only 40 epochs to train. The initial learning rate was 0.01 and then divided by 10 at the 20-th and the 30-th epochs.

Similarly to the CIFAR experiments, we used SGD optimizer with momentum of 0.9 and weight decay of $1e-4$.

### A.7 HARDWARE-ACCURACY TRADE-OFF

When operating on a single image at inference time (rather than on a large batch), the memory footprint of the weights is not negligible anymore (Mishra et al., 2017). Therefore, we need to also account for the bit-width required for storing the quantized weights. In Table 14 we report the optimal activation bit width and addition factor for each power constraint in a certain setting. Specifically, we use ZeroQ to quantize the activations of a pre-trained full-precision ResNet-50. We then measure the maximal addition factor per neuron, which defines the bit width $b_R$ required to store the weights. We can observe that overall, the increase in the runtime memory footprint of the weights is relatively low, especially in the low power regimes.

Up to now, we have only shown results with the bit width $\tilde{b}_x$ (and corresponding additions factor $R$) that is optimal in terms of classification accuracy. However, for a given power budget $P$, choosing $\tilde{b}_x$ (and $R$) can be done while also accounting for other factors, like latency and memory footprint. We illustrate this in Table 15 for the case of a power constraing corresponding to 2-bit MAC. Here, we report results for all options for $\tilde{b}_x$ and $R$ that conform to that power budget. While $\tilde{b}_x = 6$ and $R = 1.16$ is optimal in terms of classification accuracy, the user can choose other options, *e.g.*, according to latency or memory constraints.

| POWER ($b_x/b_w$) | $\tilde{b}_x$ | LATENCY($= R$) | $b_R$ | ACTIVATIONS MEMORY | WEIGHTS MEMORY |
|---|---|---|---|---|---|
| 2/2 | 6 | 1.16× | 3 | 3× | 1.5× |
| 3/3 | 6 | 2.25× | 3 | 2× | 1× |
| 4/4 | 7 | 2.9× | 3 | 1.75× | 0.75× |
| 5/5 | 8 | 3.5× | 4 | 1.6× | 0.8× |
| 6/6 | 8 | 4.75× | 5 | 1.33× | 0.83× |
| 7/7 | 8 | 6.06× | 5 | 1.14× | 0.714× |
| 8/8 | 8 | 7.5× | 5 | 1× | 0.625× |

Table 14: **Runtime memory footprint of PANN.** We report the increase in the memory required to store the weights and activations, when using PANN. Each row specifies a power budget, corresponding to a $b_x$ bit width unsigned MAC. We follow Alg. 1 to find the optimal bit width $\tilde{b}_x$ and the additions factor $R$, which is equal to the latency increase. We measure the maximal value of additions per neuron which defines the required number of bits for storing the weights ($b_R$). Then, we calculate the increase in weights memory footprint as $b_R/b_x$.

| $\tilde{b}_x$ | LATENCY($= R$) | $b_R$ | ACTIVATIONS MEMORY | WEIGHTS MEMORY | ACCURACY [%] |
|---|---|---|---|---|---|
| 2 | 4.5× | 5 | 1× | 2.5× | 0.9 |
| 3 | 2.83× | 3 | 1.5× | 1× | 6.55 |
| 4 | 2.0× | 3 | 2× | 0.75× | 61.15 |
| 5 | 1.5× | 2 | 2.5× | 0.4× | 65.69 |
| **6** | **1.16×** | **2** | **3×** | **0.33×** | **71.55** |
| 7 | 0.92× | 2 | 3.5× | 0.28× | 70.18 |
| 8 | 0.75× | 2 | 4× | 0.25× | 60.01 |

Table 15: **Hardware-accuracy trade-off.** Here we analyze the run-time memory footprint and latency increase for all different values of $\tilde{b}_x$ and $R$ that lead to the same power of a 2-bit unsigned MAC (blue curve in Figure 3(a)). For each setting we measure the classification accuracy of ResNet-50 on ImageNet. Here we use ACIQ (Banner et al., 2019) for quantizing the activations. The baseline (pure ACIQ) accuracy is 0.20% (Table 2, third column, last row).

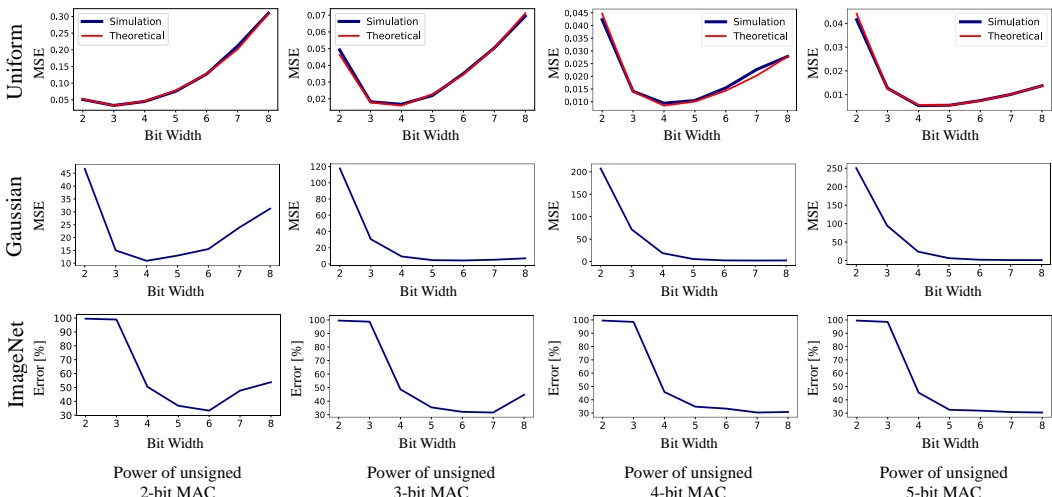

Figure 15: **Optimal bit width analysis for PANN.** The first two rows depict the MSE as a function of the activation bit width $\tilde{b}_x$ for the cases where the weights and the activations are uniformly distributed and for the setting in which they are Gaussian (the activations are further subjected to a ReLU function in this case). The third row shows the classification error of a ResNet18 model on ImageNet. Although the precise value of the optimal $\tilde{b}_x$ is a bit different than the first two rows, the qualitative behavior is similar. For both for the Gaussian simulation and the ImageNet results, we used ACIQ Banner et al. (2019) to quantize the activations.

## A.8 QUANTIZATION ERROR ANALYSIS

Figure 15 (first row) depicts $\text{MSE}_{\text{PANN}}$ (Eq. (19)) as a function of $\tilde{b}_x$ for several power budgets $P$. First, it can be seen that our theoretical analysis agrees well with simulations. Second, it shows that the optimal $\tilde{b}_x$ (where the minimum MSE is attained) increases with the power budget. This implies that at higher power budgets, it is preferable to increase the bit width of the activations on the expense of reducing the number of additions $R$. The second row of the figure illustrates that the qualitative conclusions drawn from our uniform distribution analysis also hold when the weights are Gaussian and the activations are Gaussian numbers after a ReLU function (here we used the ACIQ quantizer (Banner et al., 2019)). In the third row, we show that similar behaviors characterise the error rates of a ResNet-18 model for ImageNet classification when using PANN to quantize its weights and ACIQ to quantize the activations.

## A.9 PROOF OF EQ. (14)

Let $\boldsymbol{w}$ and $\boldsymbol{x}$ be statistically independent random vectors, each with iid components. Recall that $\boldsymbol{w}_q$ and $\boldsymbol{x}_q$ are obtained by applying a scalar function on each of the elements of $\boldsymbol{w}$ and $\boldsymbol{x}$, respectively. Therefore $\boldsymbol{w}_q$ and $\boldsymbol{x}_q$ also have iid components. We assume that $\boldsymbol{w} = \boldsymbol{w}_q + \boldsymbol{\varepsilon}_{\boldsymbol{w}}$ and $\boldsymbol{x} = \boldsymbol{x}_q + \boldsymbol{\varepsilon}_{\boldsymbol{x}}$, where $\mathbb{E}[\boldsymbol{\varepsilon}_{\boldsymbol{w}}|\boldsymbol{w}] = 0$ and $\mathbb{E}[\boldsymbol{\varepsilon}_{\boldsymbol{x}}|\boldsymbol{x}] = 0$. Then we have that

$$
\begin{aligned}
\text{MSE} &= \mathbb{E}\left[\left(\boldsymbol{w}^T\boldsymbol{x} - \boldsymbol{w}_q^T\boldsymbol{x}_q\right)^2\right] \\
&= \mathbb{E}\left[\left(\boldsymbol{w}^T\boldsymbol{x} - (\boldsymbol{w}+\boldsymbol{\varepsilon}_{\boldsymbol{w}})^T(\boldsymbol{x}+\boldsymbol{\varepsilon}_{\boldsymbol{x}})\right)^2\right] \\
&= \mathbb{E}\left[\left(\boldsymbol{w}^T\boldsymbol{x} - (\boldsymbol{w}^T\boldsymbol{x} + \boldsymbol{w}^T\boldsymbol{\varepsilon}_{\boldsymbol{x}} + \boldsymbol{\varepsilon}_{\boldsymbol{w}}^T\boldsymbol{x} + \boldsymbol{\varepsilon}_{\boldsymbol{w}}^T\boldsymbol{\varepsilon}_{\boldsymbol{x}}))\right)^2\right] \\
&= \mathbb{E}\left[\left(\boldsymbol{w}^T\boldsymbol{\varepsilon}_{\boldsymbol{x}} + \boldsymbol{\varepsilon}_{\boldsymbol{w}}^T\boldsymbol{x} + \boldsymbol{\varepsilon}_{\boldsymbol{w}}^T\boldsymbol{\varepsilon}_{\boldsymbol{x}}\right)^2\right]
\end{aligned}
\tag{21}
$$

and therefore,

$$
\begin{aligned}
\text{MSE} &= \mathbb{E}\left[\left(\boldsymbol{w}^T\boldsymbol{\varepsilon}_{\boldsymbol{x}}\right)^2\right] + \mathbb{E}\left[\left(\boldsymbol{\varepsilon}_{\boldsymbol{w}}^T\boldsymbol{x}\right)^2\right] + \mathbb{E}\left[\left(\boldsymbol{\varepsilon}_{\boldsymbol{w}}^T\boldsymbol{\varepsilon}_{\boldsymbol{x}}\right)^2\right] \\
&\quad + 2\mathbb{E}\left[\boldsymbol{w}^T\boldsymbol{\varepsilon}_{\boldsymbol{x}}\boldsymbol{\varepsilon}_{\boldsymbol{w}}^T\boldsymbol{x}\right] + 2\mathbb{E}\left[\boldsymbol{w}^T\boldsymbol{\varepsilon}_{\boldsymbol{x}}\boldsymbol{\varepsilon}_{\boldsymbol{w}}^T\boldsymbol{\varepsilon}_{\boldsymbol{x}}\right] + 2\mathbb{E}\left[\boldsymbol{\varepsilon}_{\boldsymbol{w}}^T\boldsymbol{x}\boldsymbol{\varepsilon}_{\boldsymbol{w}}^T\boldsymbol{\varepsilon}_{\boldsymbol{x}}\right].
\end{aligned}
\tag{22}
$$

We now turn to show that the last three terms equal zero. For the first of those terms, we have

$$
\begin{aligned}
\mathbb{E}\left[\boldsymbol{w}^T\boldsymbol{\varepsilon}_{\boldsymbol{x}}\boldsymbol{\varepsilon}_{\boldsymbol{w}}^T\boldsymbol{x}\right] &= \mathbb{E}\left[\boldsymbol{w}^T\boldsymbol{\varepsilon}_{\boldsymbol{x}}\boldsymbol{x}^T\boldsymbol{\varepsilon}_{\boldsymbol{w}}\right] \\
&= \mathbb{E}\left[\mathbb{E}\left[\boldsymbol{w}^T\boldsymbol{\varepsilon}_{\boldsymbol{x}}\boldsymbol{x}^T\boldsymbol{\varepsilon}_{\boldsymbol{w}} \mid \boldsymbol{w}, \boldsymbol{\varepsilon}_{\boldsymbol{w}}\right]\right] \\
&= \mathbb{E}\left[\boldsymbol{w}^T\mathbb{E}\left[\boldsymbol{\varepsilon}_{\boldsymbol{x}}\boldsymbol{x}^T \mid \boldsymbol{w}, \boldsymbol{\varepsilon}_{\boldsymbol{w}}\right]\boldsymbol{\varepsilon}_{\boldsymbol{w}}\right] \\
&= \mathbb{E}\left[\boldsymbol{w}^T\mathbb{E}\left[\boldsymbol{\varepsilon}_{\boldsymbol{x}}\boldsymbol{x}^T\right]\boldsymbol{\varepsilon}_{\boldsymbol{w}}\right] \\
&= 0,
\end{aligned}
\tag{23}
$$

where in the second line we used the law of total expectations, in the fourth line we used the fact that the pair $\{\boldsymbol{w}, \boldsymbol{\varepsilon}_{\boldsymbol{w}}\}$ is statistically independent of the pair $\{\boldsymbol{x}, \boldsymbol{\varepsilon}_{\boldsymbol{x}}\}$, and in the fifth line we used the fact that $\mathbb{E}[\boldsymbol{\varepsilon}_{\boldsymbol{x}}\boldsymbol{x}^T] = \mathbb{E}[\mathbb{E}[\boldsymbol{\varepsilon}_{\boldsymbol{x}}\boldsymbol{x}^T \mid \boldsymbol{x}]] = \mathbb{E}[\mathbb{E}[\boldsymbol{\varepsilon}_{\boldsymbol{x}}|\boldsymbol{x}]\boldsymbol{x}^T] = 0$ because of our assumption that $\mathbb{E}[\boldsymbol{\varepsilon}_{\boldsymbol{x}}|\boldsymbol{x}] = 0$.

For the second among the last three terms in (22), we have that

$$
\begin{aligned}
\mathbb{E}\left[\boldsymbol{w}^T\boldsymbol{\varepsilon}_{\boldsymbol{x}}\boldsymbol{\varepsilon}_{\boldsymbol{w}}^T\boldsymbol{\varepsilon}_{\boldsymbol{x}}\right] &= \mathbb{E}\left[\boldsymbol{\varepsilon}_{\boldsymbol{x}}^T\boldsymbol{w}\boldsymbol{\varepsilon}_{\boldsymbol{w}}^T\boldsymbol{\varepsilon}_{\boldsymbol{x}}\right] \\
&= \mathbb{E}\left[\left[\boldsymbol{\varepsilon}_{\boldsymbol{x}}^T\boldsymbol{w}\boldsymbol{\varepsilon}_{\boldsymbol{w}}^T\boldsymbol{\varepsilon}_{\boldsymbol{x}} \mid \boldsymbol{\varepsilon}_{\boldsymbol{x}}\right]\right] \\
&= \mathbb{E}\left[\boldsymbol{\varepsilon}_{\boldsymbol{x}}^T\left[\boldsymbol{w}\boldsymbol{\varepsilon}_{\boldsymbol{w}}^T \mid \boldsymbol{\varepsilon}_{\boldsymbol{x}}\right]\boldsymbol{\varepsilon}_{\boldsymbol{x}}\right] \\
&= \mathbb{E}\left[\boldsymbol{\varepsilon}_{\boldsymbol{x}}^T\left[\boldsymbol{w}\boldsymbol{\varepsilon}_{\boldsymbol{w}}^T\right]\boldsymbol{\varepsilon}_{\boldsymbol{x}}\right] \\
&= 0,
\end{aligned}
\tag{24}
$$

where we used the fact that the pair $\{\boldsymbol{w}, \boldsymbol{\varepsilon}_{\boldsymbol{w}}\}$ is independent of $\boldsymbol{\varepsilon}_{\boldsymbol{x}}$, and $\mathbb{E}[\boldsymbol{w}\boldsymbol{\varepsilon}_{\boldsymbol{w}}^T] = \mathbb{E}[\mathbb{E}[\boldsymbol{w}\boldsymbol{\varepsilon}_{\boldsymbol{w}}^T|\boldsymbol{w}]] = \mathbb{E}[\boldsymbol{w}\mathbb{E}[\boldsymbol{\varepsilon}_{\boldsymbol{w}}|\boldsymbol{w}]^T] = 0$ because of our assumption that $\mathbb{E}[\boldsymbol{\varepsilon}_{\boldsymbol{w}}|\boldsymbol{w}] = 0$.

For the the last term in (22), we have that

$$
\begin{aligned}
\mathbb{E}\left[\boldsymbol{\varepsilon}_{\boldsymbol{w}}^T\boldsymbol{x}\boldsymbol{\varepsilon}_{\boldsymbol{w}}^T\boldsymbol{\varepsilon}_{\boldsymbol{x}}\right] &= \mathbb{E}\left[\boldsymbol{\varepsilon}_{\boldsymbol{w}}^T\boldsymbol{x}\boldsymbol{\varepsilon}_{\boldsymbol{x}}^T\boldsymbol{\varepsilon}_{\boldsymbol{w}}\right] \\
&= \mathbb{E}\left[\mathbb{E}\left[\boldsymbol{\varepsilon}_{\boldsymbol{w}}^T\boldsymbol{x}\boldsymbol{\varepsilon}_{\boldsymbol{x}}^T\boldsymbol{\varepsilon}_{\boldsymbol{w}} \mid \boldsymbol{\varepsilon}_{\boldsymbol{w}}\right]\right] \\
&= \mathbb{E}\left[\boldsymbol{\varepsilon}_{\boldsymbol{w}}^T\mathbb{E}\left[\boldsymbol{x}\boldsymbol{\varepsilon}_{\boldsymbol{x}}^T \mid \boldsymbol{\varepsilon}_{\boldsymbol{w}}\right]\boldsymbol{\varepsilon}_{\boldsymbol{w}}\right] \\
&= \mathbb{E}\left[\boldsymbol{\varepsilon}_{\boldsymbol{w}}^T\mathbb{E}\left[\boldsymbol{x}\boldsymbol{\varepsilon}_{\boldsymbol{x}}^T\right]\boldsymbol{\varepsilon}_{\boldsymbol{w}}\right] \\
&= 0,
\end{aligned}
\tag{25}
$$

where we used the fact that the pair $\{x, \varepsilon_x\}$ is independent of $\varepsilon_w$, and $\mathbb{E}[x\varepsilon_x^T] = 0$, as in (23). We thus remain only with the first three terms of (22), so that

$$
\begin{aligned}
\text{MSE} &= \mathbb{E}\left[\left(w^T \varepsilon_x\right)^2\right] + \mathbb{E}\left[\left(\varepsilon_w^T x\right)^2\right] + \mathbb{E}\left[\left(\varepsilon_w^T \varepsilon_x\right)^2\right] \\
&= \mathbb{E}\left[w^T \varepsilon_x \varepsilon_x^T w\right] + \mathbb{E}\left[x^T \varepsilon_w \varepsilon_w^T x\right] + \mathbb{E}\left[\varepsilon_w^T \varepsilon_x \varepsilon_x^T \varepsilon_w\right] \\
&= \mathbb{E}\left[\mathbb{E}\left[w^T \varepsilon_x \varepsilon_x^T w \mid w\right]\right] + \mathbb{E}\left[\mathbb{E}\left[x^T \varepsilon_w \varepsilon_w^T x \mid x\right]\right] + \mathbb{E}\left[\mathbb{E}\left[\varepsilon_w^T \varepsilon_x \varepsilon_x^T \varepsilon_w \mid \varepsilon_w\right]\right] \\
&= \mathbb{E}\left[w^T \mathbb{E}\left[\varepsilon_x \varepsilon_x^T \mid w\right] w\right] + \mathbb{E}\left[x^T \mathbb{E}\left[\varepsilon_w \varepsilon_w^T \mid x\right] x\right] + \mathbb{E}\left[\varepsilon_w^T \mathbb{E}\left[\varepsilon_x \varepsilon_x^T \mid \varepsilon_w\right] \varepsilon_w\right] \\
&= \mathbb{E}\left[w^T \mathbb{E}\left[\varepsilon_x \varepsilon_x^T\right] w\right] + \mathbb{E}\left[x^T \mathbb{E}\left[\varepsilon_w \varepsilon_w^T\right] x\right] + \mathbb{E}\left[\varepsilon_w^T \mathbb{E}\left[\varepsilon_x \varepsilon_x^T\right] \varepsilon_w\right] \\
&= \mathbb{E}\left[w^T \left(\sigma_{\varepsilon_x}^2 I\right) w\right] + \mathbb{E}\left[x^T \left(\sigma_{\varepsilon_w}^2 I\right) x\right] + \mathbb{E}\left[\varepsilon_w^T \left(\sigma_{\varepsilon_x}^2 I\right) \varepsilon_w\right] \\
&= d\left(\sigma_w^2 \sigma_{\varepsilon_x}^2 + \sigma_x^2 \sigma_{\varepsilon_w}^2 + \sigma_{\varepsilon_x}^2 \sigma_{\varepsilon_w}^2\right),
\end{aligned}
\tag{26}
$$

where $\sigma_w^2, \sigma_x^2, \sigma_{\varepsilon_w}^2$, and $\sigma_{\varepsilon_x}^2$ denote the second-order moments of the elements of $w, x, \varepsilon_w$, and $\varepsilon_x$, respectively, and $I$ is the $d \times d$ identity matrix. Here, in the fifth equality we used the fact that $w$ is independent of $\varepsilon_x$, $x$ is independent of $\varepsilon_w$, and $\varepsilon_w$ is independent of $\varepsilon_x$. In the sixth equality we used the fact that $\varepsilon_x$ and $\varepsilon_w$ are iid vectors with zero mean, since $\mathbb{E}[\varepsilon_x] = \mathbb{E}[\mathbb{E}[\varepsilon_x|x]] = 0$ and $\mathbb{E}[\varepsilon_w] = \mathbb{E}[\mathbb{E}[\varepsilon_w|w]] = 0$. This completes the proof of Eq. (14) in the main text.

