# OpenReview forum: "Beyond Quantization: Power aware neural networks"
_ICLR.cc/2022/Conference — ICLR 2022 Submitted_

### Official Review · Reviewer_yuqG · 2021-10-30

**Correctness:** 3
**Technical Novelty And Significance:** 3
**Empirical Novelty And Significance:** 3
**Recommendation:** 6
**Confidence:** 2

**Main Review:**

** Strength **
1. The idea that quantized the model using the power budget is very important to real-world applications.
 2. The paper achieved superior accuracy than the previously developed quantized methods.
** Weakness **
1. The experimental application is too narrow. Since the authors mentioned that the proposed method can be applied to "any" model, a wide variety of applications (BERT for QnA task, etc..) should be included.
2. If each layer has a different quantization bit-width, the hardware implementation cost could be increased when compared to the fixed bit-width-based hardware.

** Minor Comments **
- Please fix me if I am misunderstood. The proposed method determines the quantization bit-width using the power budget of the entire network, so each layer or neuron has a different bit-width. If my understanding is right, mentioning this explicitly might help to understand the paper.
- The phrase "to quantize the weights and activations at post-training" in the caption of Figure 1 is a bit confusing. In the current explanation, the readers can be confused why removing the multiplier increases accuracy.
- In the case of batchnorm, the authors mentioned that "Batch-norm layers should first be absorbed into the weights and biases". How can it be handled when using the layernorm?
- The state-of-the-art models employed activations with negative values. How to convert to unsigned integers when using gelu or elu?
- The paper can be enhanced if the authors provide how many bits are allocated for each layer for 2 bits (power) in Table 2.
- There is a typo on page 2, "Our approach can work in combination" -> "Our approach can work in combination"

**Summary Of The Paper:**

This paper proposed a method for reducing the power consumption of deep neural networks called a power-aware neural network (PANN).  The proposed method exploits the entire power consumption of the model to determine the quantization bit widths, so it could achieve higher accuracy than some previously developed methods when the power budget is set to ultra-low bits such as 2-bit or 3-bit.

**Summary Of The Review:**

The idea of finding the optimal quantized model in terms of the power budget is an important issue for real-world applications. Although some part of the paper is ambiguous to me, I think this paper could be a bridge to reduce the gap between the model researchers and hardware engineers.

---

> ### Author Response · Authors · 2021-11-21
> **Response to reviewer yuqG**
>
> **The experimental application is too narrow.**
>
> Please note that any model whose compute is dominated by multiply-accumulate operations should benefit from PANN. In this paper we indeed focused on convolution based models for image classification, where we examined our approach on 9 different models with quite different architectures (this evaluation is significantly more comprehensive than that used in most quantization papers). We agree that language models are also an interesting use case, but we leave this for future work.
>
> **If each layer has a different quantization bit-width, the hardware implementation cost could be increased when compared to the fixed bit-width-based hardware.**
>
> Thanks! Please note that when using PANN, all layers have the same bit width. The addition factor R is chosen by Alg.1 and applied to all layers.
>
> You may be referring to the fact that R is the average number of additions per element, whereas the maximal value might change between layers. However, we take the maximal value among all elements in all layers, and use it to determine the number of bits for the weights. This bit width is used for all weights in all layers. This is the value $b_{R}$ reported in Tables 14-15.
>
> **The proposed method determines the quantization bit-width using the power budget of the entire network, so each layer or neuron has a different bit-width.**
>
> This is incorrect. We determine the power according to a baseline model which quantizes both the weights and the activations to bit width $b_{x}$. Then, under this power constraint,  we use PANN to find the optimal activation bit width $\tilde{b}_x$ and additions budget R (according to Alg.1). The activations in layers are then quantized using the same activation bit width, $\tilde{b}_x$. The weights in all layers are quantized using the same bit width (derived from R).
>
> **The phrase "to quantize the weights and activations at post-training" in the caption of Figure 1 is a bit confusing.**
>
> Thanks. We rephrased it.
>
> **In the case of batchnorm, the authors mentioned that "Batch-norm layers should first be absorbed into the weights and biases". How can it be handled when using the layernorm?**
>
> Thanks, this is an excellent point that relates to any quantization method (not only PANN).
> In batchnorm, once the training ends, the normalization parameters are fixed. Therefore, before deployment, they can be absorbed into the weights. In layernorm, however, the normalization parameters depend on the input, and therefore cannot be absorbed into the weights prior to deployment. Again, this is true for any quantization method.
>
> **The state-of-the-art models employed activations with negative values. How to convert to unsigned integers when using gelu or elu?**
>
> We can always separate the activation into positive and negative parts as well as the weights. Specifically, we will need to divide also the activation $x$ into $x^{+}$ and $x^{-}$.
> Then, we can have only unsigned multiplications:
> $Wx = \left(W^{+}x^{+}-W^{-}x^{+}\right) - \left(W^{+}x^{-}-W^{-}x^{-}\right)$.
>
> **The paper can be enhanced if the authors provide how many bits are allocated for each layer for 2 bits (power) in Table 2.**
>
> Please note that in Table 15 in the Appendix, we explicitly analyzed the case of 2-bit MAC power. As we mentioned, we assume the same activation bit-width for all layers. As for the weights, since the optimal activation bit width is 6, the corresponding R we get is 1.16 (Alg. 1). We measured the maximal value of addition per neuron over all layers of ResNet50 and found that 2-bit (numbers 0-3) is the required representation.

---

### Official Review · Reviewer_GEgt · 2021-11-02

**Correctness:** 1
**Technical Novelty And Significance:** 2
**Empirical Novelty And Significance:** 1
**Recommendation:** 3
**Confidence:** 4

**Main Review:**

Weakness:

* The authors rely on a 2000 publication to support the argument that dynamic power dominates. That's hardly definitive today, especially the power is simulated in the 5nm regime. Stronger support and/or better methodology is needed.
* The authors didn't consider the memory energy, which usually dominates the total energy, as opposed to MAC energy. In general, the experimental methodology is too simplistic, essentially a python script counting bit flips.There are just way too many other things that affect the energy. At least try some CACTI models; better yet, use a memory compiler.

**Summary Of The Paper:**

The paper proposes to remove signed arithmetics and removing multiplication. The power model is limited to dynamic power, and just the MAC power.

**Summary Of The Review:**

The idea is of intellectual merit, but the evaluation ignores too much important aspect of real energy/power modeling. The results are far from convincing in that regard.

---

> ### Author Response · Authors · 2021-11-21
> **Response to reviewer GEgt**
>
> **The authors rely on a 2000 publication to support the argument that dynamic power dominates. That's hardly definitive today, especially the power is simulated in the 5nm regime. Stronger support and/or better methodology is needed.**
>
> Thanks for this comment. Indeed, with the reduction of gate size over the years, the ratio between the dynamic power and the static power has been reduced. Nevertheless, dynamic power is still a significant source of power consumption. Following your comment, we rephrased this sentence to: “We are focusing on dynamic power, which is a prominent source of power consumption”. We also added the percentage of the dynamic and static power from the overall power, as measured on our 5nm simulations. As far as we know, this is the most advanced technology and therefore might have the highest static power. Please see Appendix A.1.1.
>
> In any case, we would like to stress that all existing approaches for efficient DNN design, focus on dynamic power (decreasing the number of MACs or bit toggling) and not on static power. This includes AdderNet, ShiftAddNet, and as well as dedicated architectures like MobileNet, Efficienet, BWN, TWN etc’, and all quantization methods. There is really little that can be done to control the static power on the DNN architecture level.
>
> **The authors didn't consider the memory energy, which usually dominates the total energy, as opposed to MAC energy.**
>
> Thanks for this great comment!
> We completely agree that the power associated with data movement can be non-negligible in certain settings. However, it strongly depends on the hardware implementation and the physical location and the type of the memory used. The ratio between the compute power and the memory movement power is not publicly available for Nvidia’s GPUs, Google’s TPUs, etc’. But the current tendency is towards processing-in-memory accelerators, which significantly reduce the power associated with memory movement. Please see the recent paper [R1], which reviews processing-in-memory accelerators. Table 1 of that paper shows that in the leading approaches, the computation is significantly more power consuming than the memory movement. For example, choosing DRAM, an 8-bit full adder requires 1.18nJ ($\approx$ 1.18/8 for 1-bit adder, denoted below as $P_{add_{1bit}}$), while reading 512 bits of data requires only 0.66nJ ($\approx$ 0.66/512 for 1-bit transfer, denoted below as $P_{trans_{1bit}}$).
> Using these numbers, here is a concrete example for the compute power and memory power of a regular 2-bit MAC vs. PANN:
>
> | Method | Compute | Memory     |
> | :---        |    :-----------      |         :-----------      |
> | Regular 2-bit MAC  | 4 * $P_{add_{1bit}}$+32 * $P_{add_{1bit}}$=5.31nJ  | 2 * $P_{trans_{1bit}}$+2 * $P_{trans_{1bit}}$= 0.005nJ   |
> | Details for Regular 2-bit MAC                            | Assuming 4 one-bit adders are required for a 2x2 bits multiplier and 32 one-bit adders for 32-bits accumulator. | Both weights and activations are quantized to 2 bits. |
> | PANN | 1.1 * 32 * $P_{add_{1bit}}$=5.192nJ |  6 * $P_{trans_{1bit}}$+3 * $P_{trans_{1bit}}$= 0.011nJ      |
> |    Details for PANN    | Assuming 32 one-bit adders for 32-bits accumulator and the addition factor is 1.1 |  With PANN we need 6 bits for representing the activations and 3 bits for representing the weights (Table 14).   |
>
>
> We can see that in both cases the memory movement power is negligible, and overall PANN consumes even slightly less power: 5.204nJ in comparison to 5.31nJ.
>
> [R1] Angizi, Shaahin, et al. "Accelerating deep neural networks in processing-in-memory platforms: Analog or digital approach?." 2019 IEEE Computer Society Annual Symposium on VLSI (ISVLSI). IEEE, 2019.

---

> > ### Author Response · Authors · 2021-11-23
> > **Please note that we updated our answer about the comment "The authors rely on a 2000 publication.."**
> >
> > In any case, we would like to stress that all existing approaches for efficient DNN design, focus on dynamic power (decreasing the number of MACs or bit toggling) and not on static power. This includes AdderNet, ShiftAddNet, and as well as dedicated architectures like MobileNet, Efficienet, BWN, TWN etc’, and all quantization methods. There is really little that can be done to control the static power on the DNN architecture level.

---

> > > ### Comment · Reviewer_GEgt · 2021-11-27
> > > **DRAM power argument is weak**
> > >
> > > My impression is that in-memory DNN usually use very different approaches for performing the MAC operation, e.g., the crossbar in analog circuit, where I am not sure if your proposed digital MAC circuit applies. The numbers you cited from Table I of the ISVLSI paper are from this paper (https://people.inf.ethz.ch/omutlu/pub/ambit-bulk-bitwise-dram_micro17.pdf), which targets specifically bulk bitwise operations, not DNNs. I am not sure if you can generalize the numbers here.

---

> > > > ### Author Response · Authors · 2021-11-29
> > > > **Thanks for the response**
> > > >
> > > > Thanks a lot for your comment.
> > > > First, we’d like to emphasize that we mentioned the in-memory processing just as one example for a setting where the memory energy is much lower than the compute energy. There are also other implementations where this is the case. Particularly, in modern implementations the memory is physically close to the ALU so that the power consumption associated with data movement to/from the memory is relatively negligible.
> > > >
> > > > As for the in-memory processing, you’re correct that the implementation is not precisely the same as our MAC model. But we believe that the digital processing-in-memory architectures behave qualitatively similarly to the MACs we analyzed (there too bit flips are a major factor in the dynamic power consumption). Please see Neural Cache [1] IV.A for the MAC procedure. In this case, PANN consumes 5.197nJ in comparison to 5.312nJ consumed by the regular 2-bit MAC (using SRAM). There are also other digital processing-in-memory architectures that implement the compute part similarly to our digital MAC [2].
> > > >
> > > > Regarding the comment that these memories are not used in DNNs, note that the ReRAM device, which is capable of performing massively parallel analog multiplication-accumulation, is used in PRIME [3], and the SRAM [1], STT-MRAM [4] and SOT-MRAM [5] are used in DNNs as well.
> > > >
> > > >
> > > >
> > > > [1] Eckert, Charles, et al. "Neural cache: Bit-serial in-cache acceleration of deep neural networks." 2018 ACM/IEEE 45th Annual International Symposium on Computer Architecture (ISCA). IEEE, 2018.
> > > > ‏
> > > >
> > > > [2] Angizi, Shaahin, et al. "Processing-in-Memory Acceleration of MAC-based Applications Using Residue Number System: A Comparative Study." Proceedings of the 2021 on Great Lakes Symposium on VLSI. 2021.
> > > > ‏
> > > >
> > > > [3] Chi, Ping, et al. "Prime: A novel processing-in-memory architecture for neural network computation in reram-based main memory." ACM SIGARCH Computer Architecture News 44.3 (2016): 27-39.
> > > > ‏
> > > >
> > > > [4] Jain, Shubham, et al. "Computing in memory with spin-transfer torque magnetic RAM." IEEE Transactions on Very Large Scale Integration (VLSI) Systems 26.3 (2017): 470-483.
> > > > ‏
> > > >
> > > > [5] Angizi, Shaahin, et al. "GraphS: A graph processing accelerator leveraging SOT-MRAM." 2019 Design, Automation & Test in Europe Conference & Exhibition (DATE). IEEE, 2019.‏

---

> > > > > ### Comment · Reviewer_GEgt · 2021-11-30
> > > > > **Justification needed**
> > > > >
> > > > > >Particularly, in modern implementations the memory is physically close to the ALU so that the power consumption associated with data movement to/from the memory is relatively negligible.
> > > > >
> > > > > Can you define what you mean by "modern implementation"? Also, can you point to specific numbers justifying your claim here?
> > > > >
> > > > > Also, when I say memory, I am not just talking about the main memory energy. The SRAM and on-chip Flip Flops also consume massive amount of energy.
> > > > >
> > > > > >the ReRAM device, which is capable of performing massively parallel analog multiplication-accumulation
> > > > >
> > > > > Is your paper about analog MAC? Does your simulation faithful model analog behaviors?

---

> > > > > > ### Author Response · Authors · 2021-11-30
> > > > > > **Thanks for the response**
> > > > > >
> > > > > > Thanks a lot for your response.
> > > > > >
> > > > > > By “modern” we mean 5nm or even 15nm. We are not aware of any publication that provides concrete numbers which show that memory energy is larger than compute energy in 5nm/15nm technologies. Can you please point us to such a reference?
> > > > > > The numbers for NVIDIA’s GPUs, Google’s TPUs, or Habana’s Gaudi, are not publicly available. To the best of our knowledge, the only reference that provides concrete numbers for a setting where memory energy is larger than compute energy is an old presentation by Mark Horowitz [1]. But it talks about 45nm technology, which is obsolete. All we can say is that an indirect evidence for the importance of the compute energy, is the massive focus companies put on this in their publications.
> > > > > >
> > > > > > Note that the FF toggles are part of our model. Also, our paper is not about analog MACs of course. We just mentioned it as a response to your comment regarding PIMs that are not dedicated to DNNs. The numbers for the analog memory are irrelevant, of course. Only the digital memories are relevant.
> > > > > >
> > > > > > Finally, we would like to stress that even in the worst case scenario (older silicon process), where the memory energy is relatively higher than the compute energy, there are many use-cases in which the user will prefer significantly higher classification accuracy (~70% in comparison to ~0% with ResNet-50. See Table 2 in the paper) on the expense of a 3x higher memory footprint.
> > > > > > In a previous response, we also mentioned the case of BERCQ with 2 bit weights and 32 bit activations (full precision activations!). Specifically, for ResNet 50, with 1.1 additions per element and 6 bit activations we get classification accuracy of 73.21% (Table 2) while BERCQ with 2 bit weights and 32 bit activations achieves only 72.4% (Table 3 in the BERCQ paper). In this setting, we reduce the power consumption of the compute, while also reducing the memory footprint (by 32/6 ~5.3).
> > > > > >
> > > > > > [1] Mark Horowitz. Energy table for 45nm process. In Stanford VLSI wiki. 2014b.

---

### Official Review · Reviewer_NEeA · 2021-11-02

**Correctness:** 2
**Technical Novelty And Significance:** 2
**Empirical Novelty And Significance:** Not applicable
**Recommendation:** 3
**Confidence:** 3

**Main Review:**


### Strengths
* Power-aware DNNs are interesting and important that merits exploration.
* The mean squared quantization error of a regular uniform quantizer and the proposed PANN was theoretically analyzed given first-order approximation, and they are further compared under the same fixed power budget.
* Low-bit networks in post-training quantization with PANN exhibits only a minor degradation in accuracy compared to floating-point counterparts.

### Weaknesses
* Although bit-flips can be considered an important source of power consumption, the proposed approach of counting bit-flips and minimizing such counts per image inference is fundamentally flawed. The design of the multiply-accumulate provides an incomplete and oversimplified description of the actual hardware realization of a DNN. For instance, the summation of $N$ values can be implemented as an adder tree instead of working on a single accumulator, and would result in a reduction of the number of additions to approximately $\log(N)$ and thus reducing the number of bit-flips. In addition, how the circuit should parallelize and/or time-share computations would have great impact on the number of bit-flips. Considering only a long multiply-accumulate chain for each layer is unrealistic and not faithful to the actual hardware designs.
* The proposed method in Section 5 to replace multipliers with repeated additions by accumulation is equal or inferior to multiplication in terms of bit-flips. Intuitively, counting bit operations, repeated addition consumes $min(O(2^N M), O(2^M N))$ for an $(N,M)$-bit multiplication, whereas multiplication would use $O(MN)$.
* Some important experimental settings (bit-level details, training epochs, optimizer, etc.) are missing. It would be better to provide such details in appendix to for reproducibility as code is not available in this submission.

### Minor
* “Our approach can can work in combination with any activation quantization method.” Duplicate “can”.

**Summary Of The Paper:**

The authors observed that the power consumption is dominated by the bit toggling at the input of the accumulator and decreasing the bit-width of only the weights or only the activations has limited benefit to reduce the the power consumed by the multiplier. The paper proposed PANN, which uses tricks such as unsigned arithmetic in CNN and implement multiplications via additions to achieve a multiplier-free and power-aware neural network. Experiments on both post-training quantization and quantization-aware training showcased better performance compared to other works under the same number of bit-flips. However, the reviewer believes there are fundamental flaws that need to be thoroughly addressed before this paper is ready for ICLR.


**Summary Of The Review:**

Although power-aware DNNs are interesting and this topic is an important research direction, this paper has some fundamental flaws that prevented the reviewer from recommending acceptance.

---

> ### Author Response · Authors · 2021-11-21
> **Response to reviewer NEeA**
>
> **Considering only a long multiply-accumulate chain for each layer is unrealistic and not faithful to the actual hardware designs.**
>
> Thanks for this great comment! In our analysis, we used the most common and general MAC model since we didn’t want to constrain to a given architecture design. However, it is worth mentioning that our proposed MAC model is a basic building block in actual hardware
> implementations (for example, the systolic array in TPUs [R1,R2,R3]).
>
> We didn’t mention it in the paper, but there is roughly a linear relation between power consumption and area on chip. Therefore, we believe that comparing a mult-add tree with an adder tree would result in the same observations in favor of our method. More specifically, assuming around $b^{2}$ adders of 1-bit in the $b$x$b$ multiplier, the power consumption (which goes linearly with the area) will be around b times larger than a $b$-bit adder with $b$ 1-bit adders. However, we leave the precise analysis of other architectures to future work.
>
> **The proposed method in Section 5 to replace multipliers with repeated additions by accumulation is equal or inferior to multiplication in terms of bit-flips. Intuitively, counting bit operations, repeated addition consumes $O(NM)$ for an $(N,M)$-bit multiplication, whereas multiplication would use $\min(O(Nlog⁡(M)),O(Mlog⁡(N)))$.**
>
> This is not true when $\min(M,N)$ is very small, as in our setting, for two main reasons.
> First, please note that we showed by Python and by real gate level simulation that the power of $(N,M)$-bit multiplication is dominated by $\left(\max(M,N)\right)^{2}$, especially in the popular Booth encoder multiplier (see Eq. (7), Fig. 9, and Appendix A.4). So, for example, 6bit x 2bit multiplication consumes roughly the same power as 6bit x 6bit multiplication. We believe this behavior has been overlooked to date, even though quantization papers often advocate using different bit widths for weights and activations.
> Second, when performing R repeated additions, one of the inputs to the accumulator remains fixed for R cycles, resulting in less bit flips. The power corresponding to implementation of MACs via repeated additions is given by Eq. (13).
> Let’s take a specific example. Assume we have 6-bit activations and 2-bit weights. Then a multiplication involves 22 bit flips (Eq. (7)) and accumulation of the multiplier output involves 18 additional bit flips on average (Eq. (4)). Namely, a regular MAC involves 40 bit flips. This is while implementing MACs using additions involves R=1.5 additions per element on average (the numbers representable by 2 bits are 0,1,2,3) and therefore involves only 12 bit flips per element (Eq. (13)).
> Of course when $\min(M,N)$ is not very small, using a multiplier can be significantly more advantageous over using repeated additions, as you note. For example if both the weights and the activations use 6 bits, then a regular MAC would still involve 40 bit flips, and our repeated additions implementation would result in 192 bit flips. Yet, this is not the setting we explore.
>
>
> **Some important experimental settings (bit-level details, training epochs, optimizer, etc.) are missing. It would be better to provide such details in appendix to for reproducibility as code is not available in this submission.**
>
> Thanks a lot, we will definitely publish the code for reproducibility of our results.
> Please note that in all our experiments we just replace the weight quantization block with PANN (eq. (12)). Therefore, we always follow the same settings as the method we compare with. Namely, we use the same activations quantization as the baseline method, as well as the same optimizer, learning rate schedule, batch size, etc’.
> We added a section where we provide the hyper-parameters used for PANN in QAT (Appendix A.6).
>
>
> **Duplicate “can”.**
>
> Thanks for noticing! we fixed it in the new revision.
>
>
> [R1] Zhang, Jeff, et al. "Thundervolt: enabling aggressive voltage underscaling and timing error resilience for energy efficient deep learning accelerators." Proceedings of the 55th Annual Design Automation Conference. 2018.
> ‏
>
> [R2] Jouppi, Norman P., et al. "In-datacenter performance analysis of a tensor processing unit." Proceedings of the 44th annual international symposium on computer architecture. 2017.
> ‏
>
> [R3] Kundu, Shamik, et al. "Toward Functional Safety of Systolic Array-Based Deep Learning Hardware Accelerators." IEEE Transactions on Very Large Scale Integration (VLSI) Systems 29.3 (2021): 485-498.‏

---

> > ### Comment · Reviewer_NEeA · 2021-11-22
> > **Thanks for the response.**
> >
> > Apologies for the incorrect comment on the problem of repeated additions, I meant to say "repeated addition consumes $min(O(N^2 M), O(M^2 N))$ for an $(N,M)$-bit multiplication, whereas multiplication would use $O(MN)$". Thus repeated addition is much worse than doing plain multiplication or with the Booth algorithm. The original review was updated to reflect this. I don't think it is true that the Booth algorithm requires $O(max(M, N)^2)$ bit-ops as stated in your rebuttal. For instance, a 6-by-2 multiplication with the 2-bit value being the multiplicand would simply reduce to a single addition of a 7-bit and a 6-bit value, using only 7 full-adders in total.

---

> > > ### Author Response · Authors · 2021-11-22
> > > **Thanks for the response**
> > >
> > > Please kindly note that the behavior of the Booth encoder we report in the paper was validated with a real gate level simulation (on a 5nm process). This conclusion is not drawn only based on our Python simulations, which count bit flips. Here is the exact cite from our paper:
> > > "We validated our observation on the $5$-nm silicon process setup. We used $8\times8$ multiplier and measured the power when one of the inputs was drawn uniformly from $[0,2^{7})$ and the other from $[0,2^{3})$. We got 95% of the power that was measured when both inputs were drawn from $[0,2^{7})$. In case of signed values, when one of the inputs was drawn uniformly from $[-2^{7},2^{7})$ and the other from $[-2^{3},2^{3})$ we observed 100% of the power that was measured when both inputs were drawn from $[-2^{7},2^{7})$.".
> > >
> > > The reason for this behavior is that the number of toggles per multiplication is affected by all inner components in the multiplier (see Fig. 1 in [R4]), many of which are not at all related to the current product, but rather to the previous one. Therefore, when having for example a sequence of MACs like -2*(-48)+ 3*(-5)+1*(51) many bits are toggled just because of the use of the 2's complement and the switching from positive to negative numbers and vice versa.
> > >
> > > Regarding "repeated addition consumes $min(O(N^2 M), O(M^2 N))$ for an $(N,M)$-bit multiplication, whereas multiplication would use $O(MN)$". Could you please specify the computation? It is unclear how repeated additions cost  $min(O(N^2 M), O(M^2 N))$.  Thanks in advance!
> > >
> > > [R4] Chang, Yen-Jen, et al. "A Low Power Radix-4 Booth Multiplier With Pre-Encoded Mechanism." IEEE Access 8 (2020): 114842-114853.

---

> > > > ### Comment · Reviewer_NEeA · 2021-11-22
> > > > **Typo in the response**
> > > >
> > > > Apologies again for the typo, I meant $min(O(2^N M), O(2^M N))$ for repeated additions for an $(N, M)$-bit multiplication, the original review has been updated for this change. Please address my concerns about this. It is unclear to me how it could be more efficient than plain multiplication that requires only $O(MN)$ bit-ops.

---

> > > > > ### Author Response · Authors · 2021-11-22
> > > > > **Thanks for the response**
> > > > >
> > > > > We completely agree that repeated additions cost $O(2^M N)$ (assuming $M$ is small and $N$ is large). This is reflected in Eq. (13), where $\tilde{b}_x$ is $N$ and $R$ is the average number of additions, which is $O(2^M)$. Namely, repeated additions cost $O(2^M N)$ (Eq. (13)), and a single multiplication costs $O(N^2)$ (Eq. (7)). But the big-$O$ notation may be deceiving because when working with very small numbers, the constants in the big-$O$ matter a lot. And again, these constants are not only functions of the current MAC. They are also functions of the previous MAC, with respect to which the bits in the internal units flip. That’s the reason we used bit flip simulations to determine the constants. So, for example if $M=2$ and $N=6$, it turns out that there’s a big advantage to using repeated additions. While if $M=6$ and $N=6$, there’s a big advantage to using a multiplier (in line with your intuition). Please refer to our original response for the exact details of this example.

---

> > > > > > ### Comment · Reviewer_NEeA · 2021-11-25
> > > > > > **Even when $M = 2$, repeated addition is inferior to multiplication.**
> > > > > >
> > > > > > Your example of $M = 2$ and $N = 6$ does not favor your argument. $M = 2$ has 4 possible values $\\{0, 1, 2, 3\\}$. Let's consider only the data paths, the maximum number of repeated additions is thus three additions of 6-bit values (at least 5 + 6 + 7 full adders, 3 half adders). Multiplication would only require a left shift of the 6-bit value (simply no-op), and an addition between a 7-bit and a 6-bit value, using only 6 full adders and 1 half adder. Even when $M = 2$, repeated addition is inferior to multiplication, and it makes no sense to scale to any larger $M$ bit-widths.

---

> > > > > > > ### Author Response · Authors · 2021-11-29
> > > > > > > **Under our settings, repeated addition is not inferior to multiplication**
> > > > > > >
> > > > > > > This is incorrect for several reasons:
> > > > > > > 1. We’re concerned with the average case, not the worst case. Although the maximum number of repeated additions is 3, the average number of required additions is smaller. For example, for ResNet 50 we only need an average of 1.1 additions per element to outperform the SoTA. Specifically, with 1.1 additions and N=6 we get 73.21% (Table 2) while BERCQ with M=2 and N=32 (full precision activations!) achieves only 72.4% (Table 3 in the BERCQ paper).
> > > > > > > 2. You’re counting how many full and half adders are employed for a given multiplication, but this is not what determines the power consumption. The power is determined by bit flips, and therefore depends also on the previous operation. Even when you compute 0 x 0 (zero times zero), all the full adders and half adders within the multiplier might toggle. This can happen if they all held the value 1 in the previous multiplication. There are also bit flips in the gates of the multiplier and bit flips in the encoder that need to be taken into account. This is why we performed the real gate level simulation, which confirmed our observation.
> > > > > > > 3. Please note that we assume a conventional NxN multiplier (even when one input has M<N bits). This is what current architectures support, because such hardware can accommodate flexible A\W bit widths (up to N). Even though when injecting an M bit number to one of the inputs of an NxN multiplier, you might expect to have less bit flips, it turns out that the average number of bit flips is the same as in NxN multiplication (observation-2 in the paper). For example, if we work with 2-bits weights, 8-bits activations and 8x8 bit multipliers, we practically consume the same power on average as with 8-bits weights and 8-bits activations. This is also exactly what our gate-level simulation shows (as we mentioned in the previous comment). And as we show, in this case it is much cheaper to use repeated additions with an accumulator than with an NxN multiplier+accumulator.
> > > > > > > 4. With PANN, we save bit toggling at one of the inputs to the accumulator because it is kept fixed for each cycle of repeated additions.

---

### Official Review · Reviewer_CvJm · 2021-11-02

**Correctness:** 3
**Technical Novelty And Significance:** 4
**Empirical Novelty And Significance:** 3
**Recommendation:** 5
**Confidence:** 4

**Main Review:**

Strong points:
* The proposed approach (PANN) is original and unlike any previous work in the field of quantization.
* The paper highlights unique new insights which are valuable for future hardware considerations.
* The proposed approach shows strong results for a given power budget for which it clearly outperforms conventional uniform quantization (under their power model assumption).
* The authors are somewhat upfront with some limitations and added the memory and latency overhead to table 2 and discussed them.
* Overall the paper is well and clearly written.

Weak points:
* The paper only considers the power consumption of the MAC array. However, the paper ignores the power consumption (and latency) required for the memory movement, which is non neglectable. The proposed approaches, both the switching to unsigned tensors (eq 5 and 6) and the switching to additions (eq 11), can further increase the memory transfer costs making this an even more important part to consider for the power model (and latency considerations).
* It is unclear whether the provided power simulations generalize to MAC operations in neural networks, and specifically of the proposed PANN approach. The simulation uses uniformly distributed random inputs (see appendix A.1), however in neural networks weights are roughly a normal distributed and similar activations (or a clipped normal distribution in the case of ReLU activations). It is unclear whether the proposed power model would also be a good approximation for such a different expected input distribution. With the proposed splitting of the positive and negative parts of the weight tensor (eq 5 and 6), the weight distribution will be even more different from the assumed uniform distribution. It is important to validate that the power model is valid under the expected distributions as most claims of the paper are based on observations from this power model.
* While the proposed approach shows good accuracy improvement for a power budget, it also has significant limitations. Namely it leads to memory overhead, latency overhead and requires dedicated hardware implementations to be utilized. For example, the 3 bit power budget model, corresponds roughly in memory and latency to a 6 bit uniform quantization model, and in this case the accuracy of the uniform model is significantly higher. Thus there is a clear trade-off between power and memory/latency.
* The assumption about uniform distributed weights and activations in section 5.3 is fairly unrealistic. Would the outcome (significantly) change under the assumption of a normal distribution?


Notes:
* Observation 1: I agree that the accumulator can have significant impact on the power usage of the MAC array. Though the required bit width for the accumulator scales with the bit width of the input tensors. The common use of 32 bit accumulator is often based on 8 bit quantization (see Rodriguez et al.), thus the example provided might overemphasis this effect.
* It is quite common to use asymmetric quantization (unsigned int + zero offset, see Jacob et al. 2018), thus unsigned integers are already frequently used. What is the advantage of the proposed splitting approach compared to standard asymmetric uniform quantization?
* Sometimes bits and power budget can be confused, especially by a less careful reader (e.g. table 2, 3). It would improve clarity if the tables and text clearly use power and only state comparable bit width where needed in brackets (now it is frequently the other way around, which on a first look seems like N bit quantization, but does not highlight the power aspect).

Questions:
* In figure 4a) (Appendix, validation of power model), the results are scaled independently for multiplication and additions. On the other side, in the paper we trade off multiplications with additions. How is this scaling exactly down and can this scaling actually have an influence on the trade-off made between multiplications and additions?
* How does PANN work during QAT? It is a bit unclear from the paper (e.g. how is the quantization simulated, is the budget set before or after training, how are ranges determined, etc)


**Summary Of The Paper:**

The paper argues that power consumption is a major obstacle in deploying DNNs to end devices and that current quantization approaches do not take power consumption directly into account and therefore are not optimal in reducing it. Using an approximate power model based on the average number of bit flips, they make two observation that are frequently overlooked by existing quantization approaches:
1) A significant portion of the power consumption of the MAC operation is due to the usage of signed integers and using unsigned integers instead can significantly reduce the power consumption.
2) The multipliers power consumption is dominated by the larger bit widths (weight or activation) and therefore using lower bit for one of the two (e.g. weights) is not power efficient.

Based on this the authors introduce a new weight quantization approach (PANN) which removes the multiply operation and replaces it with additions. This allows PANN to efficiently reduces the power consumption. For the same power budget, PANN achieves significantly higher accuracy (or effective bit width), but comes at the cost of higher latency and memory usage.


**Summary Of The Review:**

The authors propose a novel new approach to reduce directly the power consumption for inference. The originality and good results are the main strength of the paper while the weak points are mostly centered around some assumption and justifications of the power model and the limitations of the proposed approach. Overall both side are fairly in balance.

---

> ### Author Response · Authors · 2021-11-21
> **Response to reviewer CvJm (Part 1)**
>
> **The paper ignores the power consumption (and latency) required for the memory movement, which is non neglectable.**
>
> Thanks for this great comment! Regarding switching to unsigned arithmetic (eq. (5)-(6)), this doesn’t involve additional data movement from/to memory. Kindly note that we don’t change the bit width of the activations or the weights in this setting. For example, a possible implementation could employ two accumulators, one for the $W^{+}$ path and one for the $W^{-}$ path. Each weight and activation will be read once; if the weight is positive, the corresponding activation will be accumulated in the first accumulator, and if negative, in the second accumulator.
>
> Regarding replacing multiplications by additions (eq. (11)), this by itself also doesn’t involve more data movement. For example, if a weight equals 3, then we add the corresponding activation 3 times. But we read it from memory only once. Namely, the overall number of memory reads (and writes) remains the same.
>
> We do acknowledge, however, that PANN requires more memory movement because of the use of a higher bit width for the activations. In the worst-case  scenario we examined (power of 2-bit MAC), PANN uses 6-bits for the activations and 1.1 additions per element (see e.g. Table 14). So the memory movement is increased by roughly 3x because of enlarging the activation bit width by 3x. Recall that this allows obtaining e.g. 73.21% accuracy in comparison to 18.80% using BRECQ (Table 2).
>
> We completely agree that the power associated with data movement can be non-negligible in certain settings. However, it strongly depends on the hardware implementation and the physical location and the type of the memory used. The ratio between the compute power and the memory movement power is not publicly available for Nvidia’s GPUs, Google’s TPUs, etc’. But the current tendency is towards processing-in-memory accelerators, which significantly reduce the power associated with memory movement. Please see the recent paper [R1], which reviews processing-in-memory accelerators. Table 1 of that paper shows that in the leading approaches, the computation is significantly more power consuming than the memory movement. For example, choosing DRAM, an 8-bit full adder requires 1.18nJ ($\approx$ 1.18/8 for 1-bit adder, denoted below as $P_{add_{1bit}}$), while reading 512 bits of data requires only 0.66nJ ($\approx$ 0.66/512 for 1-bit transfer, denoted below as $P_{trans_{1bit}}$).
> Using these numbers, here is a concrete example for the compute power and memory power of a regular 2-bit MAC vs. PANN:
>
> | Method | Compute | Memory     |
> | :---        |    :-----------      |         :-----------      |
> | Regular 2-bit MAC  | 4 * $P_{add_{1bit}}$+32 * $P_{add_{1bit}}$=5.31nJ  | 2 * $P_{trans_{1bit}}$+2 * $P_{trans_{1bit}}$= 0.005nJ   |
> | Details for Regular 2-bit MAC                            | Assuming 4 one-bit adders are required for a 2x2 bits multiplier and 32 one-bit adders for 32-bits accumulator. | Both weights and activations are quantized to 2 bits. |
> | PANN | 1.1 * 32 * $P_{add_{1bit}}$=5.192nJ |  6 * $P_{trans_{1bit}}$+3 * $P_{trans_{1bit}}$= 0.011nJ      |
> |    Details for PANN    | Assuming 32 one-bit adders for 32-bits accumulator and the addition factor is 1.1 |  With PANN we need 6 bits for representing the activations and 3 bits for representing the weights (Table 14).   |
>
> We can see that in both cases the memory movement power is negligible, and overall PANN consumes even slightly less power: 5.204nJ in comparison to 5.31nJ.
> ‏
>
> **The simulation uses uniformly distributed random inputs (see appendix A.1), however in neural networks weights are roughly a normal distributed and similar activations (or a clipped normal distribution in the case of ReLU activations).**
>
> Please note that our observations are based on Python simulations with both uniform and Gaussian distributions (see Figures 7,8, blue crosses vs. red triangles). Also, please see Fig. 15, which shows the behavior as a function of bit width for uniform, Gaussian, and real weights/activations in a network. As can be seen, the behavior of the latter is very similar to that of the Gaussian simulation.
>
> [R1] Angizi, Shaahin, et al. "Accelerating deep neural networks in processing-in-memory platforms: Analog or digital approach?." 2019 IEEE Computer Society Annual Symposium on VLSI (ISVLSI). IEEE, 2019.

---

> > ### Author Response · Authors · 2021-11-21
> > **Response to reviewer CvJm (Part 2)**
> >
> > **The proposed approach leads to memory overhead, latency overhead and requires dedicated hardware implementations to be utilized.**
> >
> > This is true, as we discuss in the paper, but only if we do not assume any dedicated change in the hardware. Namely, the latency increases we report assumes that we remain with the same MAC model (multiplier + acc) and the same clock rate. But, as we mention in the paper, we can potentially decrease the latency if we allow changes in the hardware. For example, since PANN uses no multiplier, and since an addition is faster than multiplication (O(b) vs. O(b log(b)) [R2]), we can increase the clock rate.
> >
> > As for the 3-bit case example, please note that in the 6-bit uniform model, both the weights and the activations are quantized to 6 bits. With PANN only the activations are quantized to 6 bits. The budget of additions is R=2.25. For a fair comparison, we measured the largest addition factor per element over all layers in the network in order to see how many bits are required to store all the weights (the average is 2.25 but the maximum is naturally larger). We found that in this case, 3 bits suffice (0-7). Therefore, the memory footprint of the weights is equivalent to 3 bits and not 6 bits. Please refer to table 14. We noticed that in the first version we had a typo in this table, we apologize.
> >
> > Also, we would like to stress that with PANN we can use any bit width for the activations with any addition factor. For a fair comparison we reported only power budgets corresponding to mult-add networks with an integer bit-width. However, let us take for example the following case: when aiming for the power budget of a 4-bit MAC, the optimal bit width for the activation in PANN is 7, with an addition factor (=latency) of 2.9. However, we can also work with an addition factor R of 1.5 without significantly degrading the performance. This flexibility is a key feature of PANN: we can smoothly traverse the tradeoff between power, latency and accuracy, simply by changing the number of additions per element R. This requires no changes in the hardware.
> >
> > **The assumption about uniform distributed weights and activations in section 5.3 is fairly unrealistic. Would the outcome (significantly) change under the assumption of a normal distribution?**
> >
> > Thanks for this point. We may not have clarified it well enough. The purpose of the theoretical analysis in Section 5.3 is only to provide some insight on the differences between PANN and regular uniform quantization. We don’t use these results in Alg. 1 or in the experiments. In all our experiments we report accuracy vs. power, where the latter is computed based on the power models of Secs. 3,4.
> >
> > In Sec. 5.3, we chose to analyze the uniform case only for simplicity. Our analysis shows nice agreement to simulations on uniform data (Fig. 3(b), red and blue curves), but it also shows qualitatively similar behavior to the Gaussian case (Fig 3(b), yellow curve). Please also see Fig. 15 for the behavior of the optimal activation bit width in PANN. The behavior on ImageNet (bottom row) is indeed slightly more similar to the Gaussian case (mid row) than to the uniform case (top row). But qualitatively, all three settings are similar. We therefore believe that our theoretical analysis in Sec. 5.3 does capture the essence of the difference between PANN and RUQ, and it does provide relevant insight.
> >
> > **The common use of 32 bit accumulator is often based on 8 bit quantization (see Rodriguez et al.), thus the example provided might overemphasize the power reduction.**
> >
> > Right. The issue with using a smaller accumulator is that it reduces flexibility. Namely, it constrains us to specific quantization bit-widths.
> >
> > But following your comment we analyzed ResNet networks and chose the layer that requires the largest accumulator bit width. This corresponds to the layer with the largest value of C_in x k x k, where k is the kernel size and C_in is the number of input channels. This layer’s size is 3x3x512 (Table 1 in [R3]). We calculated the required bit width for the accumulator when the activations and weights are quantized to 2-6 bits.  We added a new paragraph “Accumulator bit width” that discusses this point. Specifically, we repeated the experiment of Fig.1 with the new bit width for the accumulator (Fig.13), and reported the power save when switching to unsigned arithmetic with the lower bit width accumulator. (Table 6). As can be seen, the save is still significant.
> > ‏
> >
> > [R2] Afshani, Peyman, et al. "Lower Bounds for Multiplication via Network Coding." 46th International Colloquium on Automata, Languages, and Programming (ICALP 2019). Schloss Dagstuhl-Leibniz-Zentrum fuer Informatik, 2019.‏
> >
> > [R3] He, Kaiming, et al. "Deep residual learning for image recognition." Proceedings of the IEEE conference on computer vision and pattern recognition. 2016.

---

> > > ### Author Response · Authors · 2021-11-21
> > > **Response to reviewer CvJm (Part 3)**
> > >
> > > **What is the advantage of the proposed splitting approach compared to standard asymmetric uniform quantization?**
> > >
> > > This is an excellent point, thanks.
> > > Asymmetric quantization uses a flexible zero point (bias), which enables quantizing an asymmetric distribution in an efficient way. This kind of quantization is usually used for activations and less for weights (which have a symmetric distribution).
> > > In any case, if used for both the weights and the activations, this method indeed requires mainly operations on non-negative (unsigned) numbers. But it increases the total number of summations. As explained in [R4,R5], after computing the unsigned MACs, we need to subtract two terms: the zero-point of the activations times the sum of the weights, and the zero-point of the weights times the sum of the activations. The sum of the weights can be precomputed, but the sum of the activations cannot. For example, for a dot product of two Nx1 vectors, our splitting approach involves N multiplications and N additions of unsigned numbers. Using asymmetric quantization, we would need N multiplications and 3N additions of unsigned numbers (or 2N additions if the sum of the weights is precomputed).
> > >
> > > **It would improve clarity if the tables and text clearly use power and only state comparable bit width where needed in brackets (now it is frequently the other way around, which on a first look seems like N bit quantization, but does not highlight the power aspect).**
> > >
> > > Thanks! We changed Tables 2-3 and Tables 7-9 according to your suggestion. Namely, replacing bits (power) with power (bits) in the first column. Please note that we further explained Table 2 and Tables 7-9 in Appendix A.5.1.
> > >
> > > **In figure 4a (Appendix, validation of power model), the results are scaled independently for multiplication and additions.**
> > >
> > > This is an important point, thanks! The scaling for the multiplier and the adder was quite similar. We updated this figure such that the same factor scales both plots (figure 4a and 4b). The scale factor was chosen such that the 5nm measurement of the multiplier coincides with the Python simulation of the multiplier at b=4. As can be seen, there is still a good agreement.
> > >
> > > **How does PANN work during QAT? It is a bit unclear from the paper (e.g. how is the quantization simulated, is the budget set before or after training, how are ranges determined, etc).**
> > >
> > > We had two types of QAT experiments:
> > > The first set of QAT experiments compared PANN against multiplication-free methods, such as AdderNet and ShiftAddNet. Since these approaches don't use MACs, rather than setting our activation bit-width using Alg. 1, we used the same activation bit-width that AdderNet and ShiftAddNet use in each comparison. So basically there is no change in memory footprint here. Regarding our addition factor, we show that even with 1 addition per element (no change in latency), we can outperform both AdderNet and ShiftAddNet with the networks and datasets that were used in the ShiftAddNet paper.
> > > In the second set of QAT experiments, we used LSQ for quantizing the activations. And similarly to post training, we replaced the weights quantization by our PANN (according to Alg.1).  We used the exact same training regime for the baseline LSQ and for our method (LSQ for quantizing the activation and PANN for quantizing the weights). The power budget is always determined before the training as described in Alg.1. For example, we define the power budget of 2 bit-width activation and weights (Eq. (3)-(4)). This is the setup that the baseline LSQ is trained with in the first and third rows of Table 3 in the paper. We added a dedicated section with all training details (Appendix A.6).
> > >
> > >
> > > [R4] Krishnamoorthi, Raghuraman. "Quantizing deep convolutional networks for efficient inference: A whitepaper." arXiv preprint arXiv:1806.08342 (2018).‏
> > >
> > > [R5] Garg, Sahaj, et al. "Confounding tradeoffs for neural network quantization." arXiv preprint arXiv:2102.06366 (2021).‏

---

### Author Response · Authors · 2021-11-21
**A new version of our paper**

We would like to thank the reviewers for their effort and the great comments and suggestions. We uploaded a revised version of our paper with changes that are detailed at each of the responses below. Thanks!

---

### Decision · Program_Chairs · 2022-01-20

**Decision:**

Reject

**Comment:**

This paper argues that the existing approaches for reducing power consumption do not model the precise power usage of each model. To remedy this an approximate power usage model is proposed using bit flips and a simple approach called PANN is introduced that relies on tricks such as unsigned arithmetics and implementation of multiplications with addition. The reviewers have found the overall direction of this paper in modeling power consumption important and have acknowledged the clarity of presentation. However, they have also raised serious concerns regarding (i) the efficacy of modeling power consumption with bit flips and ignoring memory power, (ii) its relevance to modern hardware, and (ii) the efficacy of replacing multipliers with repeated additions. Unfortunately, the paper in it is current form does not provide a compelling answer to these concerns. Given these criticisms, we don't believe that the paper is ready for publication at ICLR.